# A Keratin 12 Expression-Based Analysis of Stem-Precursor Cells and Differentiation in the Limbal–Corneal Epithelium Using Single-Cell RNA-Seq Data

**DOI:** 10.3390/biology13030145

**Published:** 2024-02-26

**Authors:** J. Mario Wolosin

**Affiliations:** Department of Ophthalmology, Black Family Stem Cell Institute and Vision Research Institute, Icahn School of Medicine at Mount Sinai, One Gustave Levy Place, New York, NY 10029, USA; jmario.wolosin@gmail.com

**Keywords:** scRNA-Seq, cornea, limbus, epithelium, Krt12, stem cells, differentiation, gene expression

## Abstract

**Simple Summary:**

The corneal epithelium protects the eye against external insults. It spreads between two domains, an outer rim or limbus, and the vision-critical central cornea. The epithelial cells constantly proliferate at the limbus from resident stem cells. The proliferating cells then migrate toward the central cornea, where there is a constant cell loss. As the cells migrate, they change their gene complexion causing changes in their properties toward the properties needed for their roles at the central cornea; this process is called differentiation. When a limbal stem cell deficiency develops, e.g., due to limbal injury, the limbus-to-cornea cell resupply stops, and the central corneal cell population dwindles. This results in corneal scarring that blocks vision. In-depth knowledge of the gene changes and the stages of change are essential for the design of corneal restorative strategies. Differentiation correlates with increases in the gene Krt12. After discovering that Krt12 increases are stepwise, this feature was exploited to identify the correlating changes in each gene of a set that included the 5647 most abundant corneal epithelial genes. The analysis yielded lists detailing the existence of hundreds to thousands of genes that change within any two sequential stages of Krt12 increase.

**Abstract:**

The corneal epithelium (CE) is spread between two domains, the outer vascularized limbus and the avascular cornea proper. Epithelial cells undergo constant migration from the limbus to the vision-critical central cornea. Coordinated with this migration, the cells undergo differentiation changes where a pool of unique stem/precursor cells at the limbus yields the mature cells that reach the corneal center. Differentiation is heralded by the expression of the corneal-specific Krt12. Processing data acquired by scRNA-Seq showed that the increase in Krt12 expression occurs in four distinct steps within the limbus, plus a single continuous increase in the cornea. Differential gene analysis demonstrated that these domains reflect discreet stages of CE differentiation and yielded extensive information of the genes undergoing down- or upregulation in the sequential transition from less to more differentiate conditions. The approach allowed the identification of multiple gene cohorts, including (a) the genes which have maximal expression in the most primitive, Krt12-negative cell cohort, which is likely to include the stem/precursor cells; (b) the sets of genes that undergo continuous increase or decrease along the whole differentiation path; and (c) the genes showing maximal positive or negative correlation with the changes in Krt12.

## 1. Introduction

The corneal epithelium (CE) constitutes the first line of defense of the vision system against pathogen invasion, and it is also a critical component of optical visual focusing. The mature CE consists of 5–6 layers of stratified non-keratinized epithelium. Its differentiation plan and homeostasis resemble that of the other outer ectoderm-derived stratified epithelia such as the epidermis and various mucosae. Briefly, the surface-exposed cells at the upper layer continuously exfoliate and this cell loss, in turn, is balanced by the proliferation of cells within the basal layer of the strata. Studies over the last four decades have shown that the short-term needs for cell replacement, whether homeostatic or acute, are fulfilled by a rapidly proliferating and partially differentiated cell type. Because the capacity for cycles of replication of these cells is limited, they have been classically referred to as transient amplifying (TA) cells. The resupply of this short-term response cell population depends on a population of stem/precursor cells which are normally in a semi-quiescent or slow cycling state but can increase their proliferation as needed to replenish a diminished TA cell population. These cells tend to reside within specialized niches and maintain an extended replicative capacity [1,2].

Another common feature of most ectoderm-derived stratified epithelia is the generalized expression of the tonofilament-forming acidic-basic cytokeratin pair Krt14 and Krt5. However, epithelia are distinguished from each other by the expression of distinct complements of acidic and basic cytokeratins, likely needed to generate tonofilaments that best suit the specific functional needs of each lining [3]. The CE is characterized by the expression of Krt12, not identified in any other tissue, and Krt3, which is rarely expressed in other stratified epithelia [4,5,6,7,8].

In addition to its cytokeratin specificity, the CE presents a unique feature that distinguishes it from all other epithelia. Owing to the avascular nature of the CE, the precursor stem cells are segregated to the vascularized outer annulus that separates the cornea from conjunctival lineages. All basal limbal cells are negative for the tissue-specific cytokeratins till the border with the transparent corneal zone and become strongly positive, in an abrupt manner, as the cell migrates into the avascular cornea or stratify within the limbus [4,9,10,11]. From a health perspective, this dual-domain arrangement represents a biological weakness. Damage to the thin limbal rim interrupts the supply of cells for the maintenance of TAs at the adjacent peripheral corneal zone and opens the door for the centripetal invasion of the pro-vascular conjunctival epithelium. This colonization leads to neovascularization, and, because the CNJE performs poorly as a fluid permeability barrier and pathogen refractor, to stromal scarring and pathogen penetration. The resulting blinding state is referred to as limbal stem cell deficiency (LSCD) [12,13]. The correction of an LSCD state requires the transfer of a suitable population of cells through a transplant operation. The success of these transplants depends on the regenerative capacity of the transferred cells [14,15]. Hence, an in-depth characterization of the precursor compartment and the stage of early differentiation are valuable tools for the design of best clinical reparative strategies and analysis of their outcomes.

Microarrays and real-time polymerase chain reactions have been used to identify gene profiles that underpin the phenotypes of lineages or specific cell cohorts of the ocular surface epithelia [16,17,18,19]. More recently, the advent of the single-cell RNA seq (scRNA) methodology has led to the genetic segregation of the CE into clusters of cells exhibiting similar GE profiles, and through this clustering, to the assignment of genes as primarily belonging to specific limbal, corneal, and conjunctival, basal, or suprabasal domains or cohorts [19,20,21,22,23]. Bearing in mind the intimate relationship between the expression of Krt12 and the differentiation process, this report uses scRNA-Seq data to examine the correlation between the levels of Krt12 expression and the expression of other genes along the limbus to the central cornea differentiation and migration axis. 

## 2. Materials and Methods

### 2.1. Tissue Procurement

Human corneas from unidentifiable cadaver donors, obtained under informed consent, were obtained from the Eye-bank from the National Research Disease Interchange (Philadelphia, PA, USA), and processed between 48 and 72 h of death. The procurement request excluded donors who had undergone chemotherapy. No donor details apart from age and sex were released. The donors were Caucasian males of ages 68, 71, and 65. The Icahn School of Medicine Institutional Review Board has determined that as per section 45 CFR Part 46 of the U.S.A. Code of Federal Regulations, unidentifiable cadaver tissue does not constitute research in human subjects (see https://grants.nih.gov/policy/humansubjects/hs-decision.htm (accessed on 7 December 2023)). The use of human tissue in this study was in accordance with the provisions of the Declaration of Helsinki.

### 2.2. Cell Preparation

Corneas were radially split into 8 segments on a cutting board. Each octile was placed on the stage of a stereoscope fitted with a rotatable black plastic board and illuminated by a 150 W Fiber Optic Dual Gooseneck Illuminator (Cole Palmer, Vernon Hills, IL, USA). After accommodating the angle of the two illuminating beans, it was possible to visualize any remaining conjunctival tissue and the limbal zone. A drop of Trypan blue was then added for 30 sec and washed with saline. The stain highlighted the freely accessible conjunctival stroma and its underlying sclera and any area of damaged or missing corneal epithelium. The conjunctiva was removed by picking up the loose tissue and cutting it out with a fine iris scissor to the edge of the scleral stain (CNJ sample, Figure 1). Next, after cutting away and discarding the blue-stained sclera, a limbal-peripheral strip (LiPe sample) was collected by a cut in the peripheral zone which resulted in a width of the peripheral tissue as close as possible to the visible width of the limbus at each particular octile. Finally, cuts were made to collect a section of the adjacent periphery (Pe sample) of a width similar to the width of the periphery included in the LiPe strip, and a segment of the central cornea showing an undamaged overlying epithelium (Co sample). The surgery was performed using the tip of Extra Keen Blue single-edged blades. The octile strips from each sample type were incubated with 2 mg/mL Dispase II (Fisher, Waltham, MA, USA) made in a bicarbonate-free 1:1 mix of Dulbecco’s modified minimum essential medium and Ham F-12 (DMEM-F12; Fisher) for 18 h, at 4 °C under a 60 tilting/min motion. At the end of this treatment, sheets of epithelial cells were either free-floating in the Dispase II solution or were lightly attached to the stroma, from which they were released by gently prodding with the tip of a jeweler’s forceps. The sheets were incubated in a 0.25% Trypsin (Fisher) solution at 37 °C for 5 min (2 mL/sample), the Trypsin medium was admixed with 4 volumes of DMEM-F12—10% fetal bovine serum (Atlanta Biologics, Flowery Branch, GA, USA), and single-cell suspensions were generated by 40 passes through a 5 mL pipette.

Cells were spun down in a clinical centrifuge and resuspended, in 2 mL of DMEM/F12, triturated again, filtered through a 70 µ filter, and cultured for 3 ½ h in a 25 mL culture flask. After visually confirming the presence of single attached cells displaying side blebs indicating incipient attachment and spreading, the flask was set horizontally for 3–4 min to allow full draining of unattached cells. The accumulated medium was fully aspirated and the lightly attached cells were released by gentle streaming from a 1 mL pipette tip. The protocol recovered about ¼ of the cells in the suspension. This brief cell adhesion protocol harvested most basal epithelial cells, while excluding or drastically minimizing the recovery of supra-basal epithelial cells.

Eighty percent of the recovered cells were used for scRNA-Seq analysis. One full set of CNJ, LiPe, Pe, and Co samples (Exp. 1) was derived from a single donor and two additional LiPe samples (Exp. 2 and 3) from another two corneas. The remaining twenty percent of the samples were complemented with 1 µg/mL propidium iodide (PI) and the forward light scattering (FSC), a relative measure of overall cell size, of the PI-negative cells was used to determine the fraction of Li and Pe derived cells within the population.

### 2.3. Single-Cell RNA-Seq Reading

Libraries were prepared using the 10x Genomics Chromium Controller (Pleasanton, CA, USA) in conjunction with the single-cell 3′ v2 kit. Briefly, the cDNA synthesis, barcoding, and library preparation were carried out according to the manufacturer’s instructions. Libraries were sequenced on a Novaseq 6000 instrument. (Illumina, San Diego, CA, USA). Sample demultiplexing, barcode processing, and unique molecular identifier (UMI) counting were performed by using the 10x Genomics pipeline Cell Ranger v.2.1.0 with default parameters. Specifically, for each library, raw reads were demultiplexed using the pipeline command ‘cell ranger mkfastq’ in conjunction with ‘bcl2fastq’ (v2.17.1.14, Illumina) to produce two fastq files: the read 1 file contains 26 bp reads, each consists of a cell barcode and a unique molecule identifier (UMI), and the read 2 file contains 96 bp reads including cDNA sequences. Reads then were aligned to the human reference genome (GRCh38), filtered, and counted using ‘cell ranger count’ to generate the gene barcode matrix. The genomics coverage ranged between 94.2 and 96.1% for all the samples. CSV files were derived from the feature barcode matrix data following the 10x Genomics instructions. The CSV files were opened in Microsoft Excel and processed as described below.

## 3. Results

### 3.1. Data Preprocessing

Figure 1B shows the flow cytometry FSC distributions for all three CE samples submitted for scRNA-Seq. Limbal basal cells are remarkably and uniformly smaller than the adjacent peripheral cells. The size transition occurs sharply at the Li–Pe interphase [24]. Suprabasal limbal cells are also much larger than their basal counterparts [20]. Basal limbal cells accounted for 53% of the cells in the sample. The percentages for the other two LiPe samples were 62% and 58%, respectively. PI-stained cells amounted to less than 2% in all cases.

In a first step, the samples were purged of non-CE lineage cells. All columns containing the melanocyte markers DCT, Tyr, Tyrp1, and/or Mlan-A (6.1% of the total; Table 1) were deleted. About 20% of the melanocyte-containing markers occurred in columns concurrently expressing keratin epithelial markers (Krt5, Krt14, or Krt12). Hence, it is reasonable to assume that these columns consist of epithelial-melanocyte hetero-cellular duplets [25]. Consistent with the expected localization of melanocytes in the limbus, there were few cells expressing melanocyte markers in the pure peripheral or central corneal samples. A few cells expressing the AQP5 CNJE marker [16] or suspected blood-exclusive CD markers were also excluded. The remaining cells expressed at least one of the two cytokeratins, Krt5 and Krt14, which are expressed in all stratified epithelia. The starting cell number, the number and percent of purged cells, and the number of cells remaining after the cell exclusion for each sample are detailed in Table 1.

Next, the mean gene count (MGC) per cell was calculated and plotted in decreasing order (Figure 2A). The resulting curves easily revealed an inflection at around an MGC of 0.5 in all cases. This accelerated count decrease is suggestive of cells in a decaying state. Thus, the analysis was limited to cells with MGCs above this value. In addition, all cells with an MGC larger than 2.5 were excluded, as those are likely to include a large percentage of cell duplets. Post-trimming images are shown in a comparative format in Figure 2B. The columns were then normalized to an MGC of 1. All the calculations were performed in a ThinkStation computer controlled by an i9 8-core processor.

### 3.2. Domain Identification

The non-CE cell, outlier purged and expression normalized spreadsheets were used to examine the distribution of Krt12. Figure 2C depicts the range of Krt12 expression in the LiPe, Pe, and Co samples for Exp. 1. Figure 2D incorporates the ranges for three LiPe samples each derived from a different donor cornea and processed at different times. The high similarity of the Krt12 plots for all three LiPe samples demonstrates the reproducibility of the scRNA-Seq protocol. Plots of all three LiPe samples in the lowest 0.5% of the full expression range (0–20 A.U.s) highlighted the presence of 5 distinct domains of Krt12 increase, D0 through D4, in all three samples. Figure 2E shows the distribution for the larger LiPe sample, LiPe-1. The pattern of Krt12 increase and the expression ranges were remarkably similar in all three LiPe samples, allowing their amalgamation into a single file. As described below, this amalgamation was essential to augment the number of differentially expressed genes (DEGs). The Krt12 changes for the amalgamated files are depicted in Figure 2F. The Krt12 range and the cell number included in each domain are displayed in the inserted table. Additionally, the CNJ, LiPe, and its subdomains, Pe, and Co domains are graphically depicted in Figure 1C, where the red color intensity, or lack of it, qualitatively represents the levels of Krt12 expression in the domain.

In contrast with the Krt12 distribution in the LiPe samples, where 32% of the cells did not express Krt12, the Pe and Co samples contained no Krt12-negative cells, and their Krt12 plots displayed a rapid rise starting from the lowest values (Figure 2E). In the Pe sample, only 20 cells, or about 1% of the total population, expressed Krt12 matching the expression range of D0–D3, 0–5.32 arbitrary units (A.U.) (Figure 2E). Thus, it can be reasonably concluded that the overwhelming majority of cells within the D0–D3 range derive from the limbus. They accounted for 60% of the cells in the amalgamated sample (4003 out of 6667), roughly consistent with the percent of Li cells observed in the flow cytometry plot (Figure 1).

### 3.3. Quality Controls and Validations

The results of this study depend on the validity of the underpinning hypothesis, that there is a close relationship between Krt12 expression level and the degree of differentiation. The results also depend on adequate genomic coverage achieved and the cell health and proper normalization of the cell sets. Concerning genomic coverage, the LiPe-1 sample of the scRNA Seq protocol identified 21,944 genes, or 65 percent of the 33,531 probed genes. This percentage is within the percent of probed genes displayed by a wide variety of organ and cell lines [26]. All the other samples displayed a very similar global gene distribution. To probe the cell health and proper normalization, it was reasoned that within the single CE, lineage cells could be expected to globally maintain a very similar number of healthy mitochondria and ribosomes. Table 2 lists the mean ± SD of the ratios between any two adjacent domains and between the first (D0) and last (D4i.2) domains, for the protein-coding mitochondrial genes (MT-) and for the genes coding for the proteins that build the small (RPS) and large (RPL) ribosomal complexes. For these comparisons, the D1, D2, and D3 domains have been amalgamated into a single D123 domain because, for most practical reasons, as shown below, they can be taken as one domain. Figure 3 shows the distribution of ratios for all genes included in each gene family. Overall, the patterns displayed support the claim that all the genes in these three sets maintain a constant rate of expression along all the defined Krt12 expression ranges, indicating that both bioenergetic function and protein biosynthesis capacity are fairly well preserved across the whole dataset. Only in the case of RPL, there may be a statistically significant, though very small, expression decrease. Additionally, the noisier D4i.1–D4i.2 comparison for RPS genes includes one R equal to a 1.34 GE decrease and one 1/R = 1.35 (R = 0.741) GE increase. Since there is little if any indication of a statistical difference in expression between these domains, to avoid multiple false positives, it will be prudent to limit the DEG definition to those genes complying with a BHp < 0.01 and a ratio higher than 1.35 or lower than 0.741.

In addition to the analysis of global gene sets, selected genes for which it is possible to formulate a priori expectations were also examined. Regarding genes that may increase with differentiation, three prospective genes code for aldehyde-converting enzymes, ALDH3A1, ALDH1A1, and TKT, which are highly expressed in the cornea [27,28]. Their very high concentration has been hypothesized to help with the transparency of the CE cells, in similarity to their role as crystallins for the ocular lens. Another gene with ample evidence for a differentiation-dependent gene is GJA1/connexin 43 [29,30,31]. Finally, an obvious choice for inclusion in this evaluation is Krt3, the other corneal-specific cytokeratin. For genes that could be expected to remain invariant, the selection included (a) EEFA1 and E1F1, two highly expressed genes that play critical but distinct roles in the efficiency and/or accuracy of translation [32,33], and the genes commonly chosen as constancy controls, ACTG1, ACTB, and GAPDH.

Figure 4A,B depicts the expression levels of all the expected correlating and invariant genes, respectively, in relationship to Krt12. In general lines, Figure 4A displays the expected results; all five genes follow the general trend of change for Krt12. Of notice, the analysis yielded an absence of statistical Krt3 and GJA1 between the D0 and D123 expression. However, these genes have very low expression in these two domains, and as shown below, the current expression matrix is not large enough to generate accurate BHp values for low-expression genes. Figure 4B presents some intriguing results. The ribosomal-associated genes EEF1A and EIF1 show only small, not statistically significant decreases very similar to those described above for the ribosomal gene sets. The genes most commonly used as control genes, though, display intriguing patterns. The actin genes undergo marked decreases with the transition from the limbal to the peripheral cornea (i.e., D123 to D4i.1). The near identity of the changes for both actin genes provides reasonable assurances of the *bona fide* nature of these expression shifts. GAPDH changes in the opposite direction. The driving forces and functional significance of these changes remain to be investigated.

### 3.4. Gene Expression Differential Analysis

Differentially expressed genes (DEGs) were defined as genes with Benjamini–Hoechsberg-adjusted *p*-values (BHp values) lower than 0.01 in an expression comparison. Due to the high frequency of cells showing nil expression for the low-expression genes, the identification of bona fide statistically significant genes decreased with the decrease in gene expression level. This is exemplified in Figure 5, which examines the relationship between gene expression and the frequency of BHp < 0.01 values in a D0 to D4 comparison. Likewise, the size of the populations being compared can be expected to have a strong influence on the calculation of *p*-values. This expectation was confirmed by comparing the BHp yields when only half of the D0 and D4 domains (e.g., by excluding even columns) were compared with the yield when using the full population sets; the identified gene lists showed identical D0/D4 expression ratios, but the BHp values for these two lists were two more than 2 orders of magnitude larger, setting a large number of genes with BHp values outside the BHp > 0.01 DEG limit. These features represent the main drawback of the methodology implemented in this study; extending the analysis to a larger set of low-expression genes, many of which might contribute to the cell phenotype will require a much larger cell base than the one used here. Hence, comparisons were made using the amalgamated file and were limited to the highest 6647 expressing genes out of the more than 21,000 genes identified by the scRNA Seq.

The domains indicated by the graphing of Krt12 rise in Figure 2 were used to establish the gene-to-Krt12 expression correlations with the following alterations. First, in the D1–D2 and D2–D3 comparisons, the 10 last and first 10 cells of each domain were excluded to avoid any difference degrading the effect of a zone where the two domains may intermingle. A D4i.1 domain was defined as the first 1000 cells of D4i, after the exclusion of the first 100 cells post-D3 domain. The D4i.2 domain consisted of the next 1000 cells. This domain trimming is indicated in Figure 2F by gaps in the Krt12 line. The D4ex subdomain was not used in the analyses. The Figure 2F inset provides an account of the range of Krt12 expression in A.U., the start and end of each domain, and the number of cells present in each (#). Table 3 provides an account of the identified DEGs for various domain comparisons, without or with the limitation to ratios exceeding 1.35-fold.

The D0–D1 comparison is particularly intriguing because it represents the expression changes associated with the initial gene expression of Krt12 that occur while cells are located within the limbal domain; it identifies the genes associated with the cell cohort enriched in the lineage stem cells. There were nearly twice as many downregulated genes as upregulated ones. Table 4 lists the top downregulated (D0/D1; R > 1) and upregulated (D1/D0; 1/R > 1) DEGs, sorted according to decreasing R or 1/R, respectively. The use of the signal ratios seems more relevant than the use of the BHp values because, as discussed above, the latter parameter is heavily influenced by the level of gene expression.

The next sequential comparisons, D1–D2 and D2–D3, yielded no DEGs. The combination of D2 and D3, to form the D23 domain, where the maximal Krt12 is three times as large as the maximal Krt12 in D1, yielded only eight upregulated DEGs exceeding the 1/R > 1.35× threshold. One of these genes, CRTAC1, was present within the upregulated D0–D1 DEG list. But, CRTAC1 and all other seven genes, namely, TGFBI, CPXM2, CLU, IGFBP7, ALDH1A1, IGFBP2, and FTH1, were within the top genes in the upregulated D123–Di4.1 list, suggesting that the change in the Krt12 rate of increase that establishes the D2 and/or D3 domains represents the earliest cell changes associated with the transition toward the gene composition of the corneal periphery cell. The comparison of D0 against the full set of limbal domain Krt12-positive cells (D123) expanded the DEG list by 30–40% from the number of DEGs yielded by the D0–D1 comparison. However, for the gene presented in both lists, the ratios for both down- and upregulated DEGs were statistically identical (Mean ± SD of 1.00 ± 0.04 and 1.00 ± 0.04 for dwon and upregulated, respectively and *p* > 0.6 for both). Thus, notwithstanding the eight genes mentioned above, most of the increase in the DEG list when using D123 instead of D1 in the comparison with D0 is likely due to the effect of the sample size described above in the *p* values.

Rationally, the next sequential comparison should be between D3 and the first Di4 subdomain Di4.1. However, given the minimal differences between D1, D2, and D3 and considering the small size of D3 and the strong effect of the population size on the chances for identifying DEGs, comparing the whole D123 with Di4.1 instead seemed a more efficient way for DEG identification. Consistent with the concept of a sudden gene expression pattern change at the Li to Pe transition, this comparison yielded very large down- and upregulated DEGs (Table 3). More than 40% of the total cell population displayed a BHp > 0.01. The top downregulated and upregulated DEGs are presented in Table 5. The comparison between D4i.1 and D4i.2 identified genes that undergo statistically significant change along with the increase in Krt12 GE. Table 6 lists the top genes of each category. The average expression values for Krt12 for the D4i.1 and D4i.2 were 37.1 and 127.3 A.U., respectively, equivalent to a 1/R of 3.43. Using this value as a basis for comparison, only three genes displayed an upregulation of larger magnitude than Krt12. In contrast, 25 genes downregulated at a faster speed than the rate of Krt12 increase. The complete set of data used to calculate the is listed in the Appendix A. 

Having identified the DEGs associated with each transition, it was possible to identify DEGs that either increase or decrease in expression at each transition and those that display a differential expression only at one of the Krt12-defined domains. The continuous down- and upregulation lists consisted of 151 and 72 genes, respectively. The top genes of these two categories are listed in the upper panels of Table 7. Most of the continuously upregulated genes are well-known genes contributing to the corneal phenotype. In addition, 179 genes undergo downregulation only at the start of Krt12 expression in D1, of which 21 had an R exceeding 1.35, the limit used to screen out false positives. These genes are strong candidates as genes that contribute to the stem/precursor cell phenotype and, thus, deserve future attention. Finally, there were 53 genes showing upregulation only in the D0–D1, but the increases were tenuous; no gene showed an increase over the 1.35 limit.

Finally, we used the single large 4700-cell Co (central cornea) sample to identify genes associated with the expression of high levels of Krt12 in the basal central corneal cells. The cell set was divided into four quartiles and DEGs between the first (Q^start^) and fourth (Q^end^) quartiles were calculated. The total number of DEGs are listed in Table 3 (Q^start^/Q^end^); the top down-, and upregulated genes are shown in Table 8. The Q^start^/Q^end^ ratio for Krt12 was 4.22. Thus, all 160 genes listed in Table 8 undergo fold changes that exceed the change in Krt12.

### 3.5. Gene Ontology Differential Analysis

From the limbus. These cells are more likely to be approaching stratification than the limbal and peripheral cells. DEGs were evinced from a comparison of the lowest 1000 Krt12 (C0-Q^start^) expression level cells vs. the highest 1000 (Co-Q^end^). The larger dataset allowed the use of 7300 genes. The comparison identified 2002 downregulated and 4277 upregulated DEGs, i.e., 86 % of the genes in the sample. Table 8 lists the top.

To explore the functional and phenotypic source of the differential gene expression between domains, a differential gene ontology analysis was implemented. The down- and upregulated whole gene lists for the D0–D1 and the D123–D4i1 sets were submitted for Panther (ver. 18.0) classification at https://www.geneontology.org/ (accessed on 6 November 2023). The resulting statistically significant domains overrepresented GO terms unique to either each of the domains were identified. Table 9’s upper left and right frames list the top unique GO terms (UGOTs) of D0 not present in D1, and the respective D1 UGOTs, limited to those with a false discovery rate < 0.01 and ranked according to the extent of gene overrepresentation (FE). Table 9’s bottom left and right frames list the top unique D123 and D4i.1, respectively. Due to the hierarchical structure of the gene ontology classification, lists contain multiple terms related to a single metabolic or bioenergetic activity. Thus, to allow the inclusion of terms associated with different functions within the list length limitation, we have selectively removed the general category terms. Four intriguing terms within D123, the domain representing all Krt12-positive cells within the limbus, are the positive regulation of protein localization to the Cajal body, maturation of LSU-rRNA, negative regulation of stem cell differentiation, and regulation of stem cell population maintenance. For the Q4i.1 set, it is possible to note lumen acidification of several organelles, desmosome organization, cellular response to arsenious substances, and glycolytic processes.

### 3.6. Gene Expression Correlation Analysis

An alternative inquiry on gene expression within the cornea can be based on the degree of correlation between the change in any gene in comparison with the change in Krt12. Due to the very shallow Krt12 rate of change, this approach is not effective for the small Krt12 increase within the limbal D0–D3 domains. The larger cell list for the central cornea, in contrast, provided a basis for obtaining robust data. The first 4600 cells of the Co sample were organized in ascending Krt12 expression levels and split into twenty 230-cell quantiles (Q1–Q20), and the correlation of each gene expression change with the changes in Krt12 was calculated using Excel’s Correl function. The top genes positively and negatively correlating with the Krt12 expression changes, respectively, are listed in the upper frame of Table 10, top frame. Table 10, lower frame lists the genes that show CC values in excess + 0.90 in both the D4 domain of LiPe, the Pe, and the Co samples.

The 2000-cell D4i subdomain was divided into 20,100-cell quantiles, and the average keratin expressions were calculated and plotted from Q1 to Q20. As depicted in Figure 6, four of the keratins displayed meaningful change. Krt3-expression increases correlated well with the increases in Krt12, though, with a lag in the response. The universal cytokeratins Krt14 and Krt5 displayed opposite changes; Krt5 changes tracked the Krt12 increases, whereas Krt14 exhibited a gradual continuous decrease. Finally, Krt75, a low-expression cytokeratin, showed a particularly intriguing distribution. Within the D0–D3 zone, it remained nearly constant (*p* > 0.05; Figure 4 inset) but decreased sharply in D4i.

### 3.7. Cytokeratin Expression

Each stratified epithelium expresses a unique set of tonofilament-forming cytokeratin pairs. This pattern is likely to represent an evolutionary adaptation to optimize each lineage to function in its specific environment. In the CE, the tissue-specific cytokeratins are Krt12 and Krt3. However, the CE expresses a large complement of other cytokeratins, in particular, the universal stratified epithelia Krt5 and Krt14. If cytokeratin expression is related to the optimization of function and the expression of Krt12 changes with the degree of differentiation, the question arises as to whether the expression of these other tonofilament-forming proteins undergoes differentiation-associated changes, and if so, how the changes relate to the change in Krt12. The gathered data present a unique opportunity to examine this issue. The 6647-gene LiPe list contains 15 cytokeratins within D4.

## 4. Discussion

ScRNA measurements are usually processed by clustering algorithms that calculate the similarity/dissimilarity of data points. One of the main uses of this methodology is the identification and characterization of heretofore undetected phenotypically distinct minute cell populations residing within an organ or tissue, e.g., immunosurveillance or incipient transformed cells. In the ocular surface, several studies have employed this methodology to assign cells as primary belonging to one of the various distinguishable domains of conjunctival and corneal lineages [20,21,22,23]. Following this clustering, using differential analysis, it becomes possible to associate each expressed gene with a specific cluster and calculate a statistical probability of the assignment. Given its significance in the management of limbal stem cell deficiency, the cluster of basal limbal cells, the site of the lineage stem/precursor cells is of particular interest.

The present study uses an alternative approach for the identification of corneal epithelial domains. It is based on the fact that the CE cells undergo a single linear differentiation path, coupled with the hypothesis that the degree of corneal epithelial cell differentiation within the basal cell compartment is reflected in the level of Krt12 expression. The resulting extensive differential expression lists and GO terms between the identified domains bear out the validity of the approach.

A graphic analysis of the rate of change of Krt12 in the LIPe sample led to the identification of four distinct Krt12 expression domains within the limbal zone. Approximately half of the cells were Krt12-negative and the rest showed Krt12 increasing in three discreet steps. These data are the first evidence for global basal limbal cell subsets. The highest Krt12 level in the limbal domains, though, amounts to well less than 1% of the maximal expression level in the central corneal epithelium. This very low expression is unlikely to translate into detectable protein levels. Nevertheless, comparative analysis of gene expression in the D0 vs. D1 domains revealed that at the start of intra-limbal Krt12 expression, a very large number of genes undergo down- or upregulation within the basal limbus. The *bona fide* stem/precursor cells can be reasonably expected to reside within D0. This proposition seems supported by the D0 UGOTs (Table 9). Firstly, regulation of hemopoiesis and embryonic organ development are direct indicators of a relationship to stemness. Secondly, the highest overrepresented term, the positive regulation of core promoter binding combined with an overrepresentation of genes involved in the negative regulation of DNA-templated transcription and the negative regulation of the RNA biosynthetic process and transcription by RNA polymerase II, yields a picture of cells with high potential for wide gene expression capacity which is prevented from strong expression by mechanisms aimed at slowing RNA and protein synthesis; these are features expected from the quiescent stem/precursor cell. Thirdly, a unique ability to deal with misfolded protein is an expected critical capacity of cells surviving in the tissue for an extended period. The NADH to ubiquinone aerobic electron transport chain UGOT may reflect a higher preference for dependence on oxygen by cells closely associated with the blood circulations than in the D1 cells, which initiate the differentiation toward the anaerobic-preferring feature of the avascular central cornea. The main D1 UGOTs, regulation of plasma membrane organization, intermediate filament cytoskeleton organization, cell–cell junction organization, and keratinocyte differentiation, are indicative of the structural changes in the differentiated phenotype, including a substantial increase in cell size.

The smaller D2 and D3 domains showed very little difference with the D1 domain; they seem to belong to cells that are mostly unchanged in gene complexation from D1. However, the subtle upsurge in these cells of the high expression genes that undergo strong increases as cells undergo a frank transition from the limbal to the periphery zones, indicates that, while still within the limbus by the Krt12 expression criterium, these cells undergo the earliest changes associated with the peripheral-corneal phenotype. It is likely that had the cell number available been larger, more genes would have fallen within the BHp < 0.01 which defines a significant difference in this study. To identify the genes associated with the cell transition from the vascular limbus to the avascular cornea, we compared the whole D123 set against D4i.1, the first 1000 cells of D4 after exclusion of the 100 cells in the transition zone. Consistent with the multiple visible sharp phenotypic changes in the CE cell at the Li–Pe interface, the D123–D4i.1 comparison yielded a very large list of DEGs including about 40% of the probed genes (Table 2).

Regarding the D123–D4i.1 comparison itself, a full interpretation of the UGOTs listed in Table 8 is not easily accomplished. From the top UGOTs for the combined D123 domains, it is clear, though, that abundant RNA processing in the nuclear Cajal body that occurs in the Li ceases or drastically decreases once cells migrate to the Pe. The Cajal body function may be associated with other UGOTs, including regulation of telomere maintenance via telomerase, since the enzyme mRNA has been found to associate with the Cajal body and its telomere length regulation [34,35]. Other intriguing UGOTs, potentially reflecting the much less differentiated state of the D123 domain in comparison to D4i.1 are the regulation of hematopoietic progenitor cell differentiation, the regulation of stem cell differentiation, and the regulation of stem cell population maintenance (all these terms are present in both the D0 and D1, and thus do not show up in either the D0 or D1 Table 8 UGOTs). The D4i.1 UGOT list includes processes that increase organelle acidification and cellular response to arsenic-containing substances. Arsenic inhibits various mitochondrial enzymes leading to the uncoupling of oxidative phosphorylation. Thus, both the acidification and the arsenic response terms may simply reflect the genetic changes associated with the transition of the cornea cell from an aerobic to an anaerobic-able gene configuration.

After completing the determination of DEGs associated with each transition, it was possible to categorize genes as continuously changing along the differentiation path or identify genes with selective differential expression at certain stages. This examination identified as many genes undergoing downregulation as upregulation. The latter cohort contains very well-recognized genes associated with the corneal phenotype, such as NQO1 and the aldehyde dehydrogenases, both important detoxification genes. How the strong downregulation of many genes contributes to the limbal–corneal phenotype remains to be examined.

Probably the most intriguing cohort, though, is that of the genes that only undergo downregulation with the transition from Do–D1; they are likely to be critical genes for the stem/precursor cell or its survival. The top two genes in the list are NR2F2 and ID3. The first is a retinoid-responsive nuclear factor with a wide gene promotion pattern. NR2F2 has been shown to act as a promoter of stemness in the epidermis [36]. ID3 is a repressor of basic helix-loop-helix transcription factors and has been shown to support human embryonic stem cell maintenance [37]. Both are particularly interesting subjects for further research. The additional two comparisons identified either the genes associated with CE cell maturation in relatively early stages in the peripheral zone next to the limbus (D4i.1–D4i.2 comparison; Table 7) or between the lower and higher extremes of Krt12 expression in the central cornea (CoQ^start^-CoQ^end^; Table 8), respectively.

An interesting global feature of the Krt12-linked GE changes can be gleaned from the ratio between the down- and upregulated genes in the various interdomain comparisons (Table 3). Within the LiPe sample, the ratio decreases as differentiation proceeds, equaling 1.68, 1.47, and 1.11 for the D0–D123, D123–D4i.1, and D4i.1–D4i.2 comparisons. Since the number of transcripts is the same for all cells in the normalized data, the implication is that gene diversity is been progressively lost as differentiation progresses, in particular within the limbal zone itself. The pattern, though, reverts at the center of the cornea, probably because the downregulation has already been mostly completed there, as suggested by the GE pattern described in Figure 7.

Two clustering-based studies of scRNA-Seq data found that the top downregulated gene in the D4i.1 to D4i.2 comparison, GPHA2, is acutely localized to a less differentiated subset of limbal cells [21,22]. In this study, though, GPHA2 was substantially expressed in the lower subdomain of the Pe. Furthermore, a D0–D4 plot for GPHA2 showed a distribution similar to that shown in Figure 4 for Krt75. Since Krt75 belongs in the same Table 8 list as GPHA2, it was intriguing to examine the distribution of these two genes in the 20 quantiles of D4 used in Figure 6 and the relationship to other downregulating genes of the D4i.1–D4i.2 comparison. Figure 7 describes the changes in GPHA2 and for the four genes with the highest correlation coefficient (>0.95) to its distribution. The inset shows the combination of the previously defined D0–D3 domains with the first four quantiles of D4 for GPHA2 Krt12 and Krt3. It is clear that in our analysis, GPHA2 achieves its maximum expression at the very start of the cell transition to the Pe domain (D4Q1). The other genes in Figure 7 display a very similar domain distribution pattern. The apparent discrepancy between both approaches remains to be resolved.

In summary, the present study demonstrates that the application of Krt12 expression as a gauge of the extent of differentiation is an efficient approach for the identification of the dynamics of gene expression changes underpinning the stem/precursor cell phenotype and the progress of CE differentiation. The Discussion provides a very limited example of the analytical possibilities afforded by the results. A future focus on individual genes may help establish a full representation of the coordination of growth and differentiation in the limbal–corneal epithelial lineage.

## 5. Conclusions

The degree of expression of Krt12, the corneal specific cytokeratin, in corneal epithelial basal cells, subjected to single-cell scRNA sequence measurement identifies five different domains characterized by the rate of Krt12 expression increase. Differential gene analysis between these domains demonstrates that they represent defined stages of differentiation. Four of these stages occur within the limbal zone, the seat of the limbal–corneal stem/precursor cells. These results combined with a study of the Krt12 gene correlation generate a whole picture of gene dynamics during all stages of differentiation.

## Figures and Tables

**Figure 1 biology-13-00145-f001:**
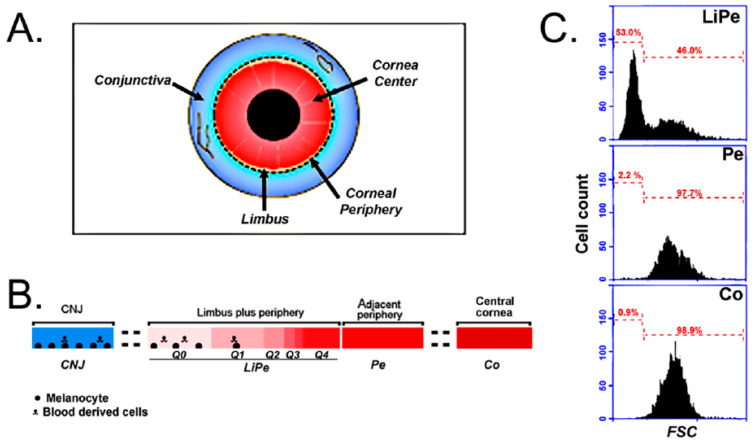
Ocular surface tissues, surgical segmentation, and the determination of limbal basal cell content and sub-limbal domains based on Krt12 expression changes. (**A**) Graphical representation of the corneal surface domains. (**B**) Flow cytometry forward light scattering (FSC) of the adherent epithelial cells collected after a 3 ½ h culture of epithelial cells harvested from either a limbal-peripheral combined zone (LiPe), an adjacent peripheral zone (Pe), or a central corneal zone (Co). FSC is a relative measure of cell size. Note that the cells collected from the LiPe sample contain similar amounts of size high and low SFC cells, whereas the Pe and Co populations consist only of high FSC cells. (**C**) Graphical representation of the four epithelial samples subjected to the scRNA seq analysis. D0–D4 represent the five subsections within the LiPe population identified by the changes in Krt12 expression. The intensity of the red color and the size of each domain has been drawn as a qualitative representation of the Krt12 level and size of each domain, respectively. The potential presence of non-epithelial cells, namely, melanocytes and blood-derived cells within the conjunctival and LiPe domains is indicated.

**Figure 2 biology-13-00145-f002:**
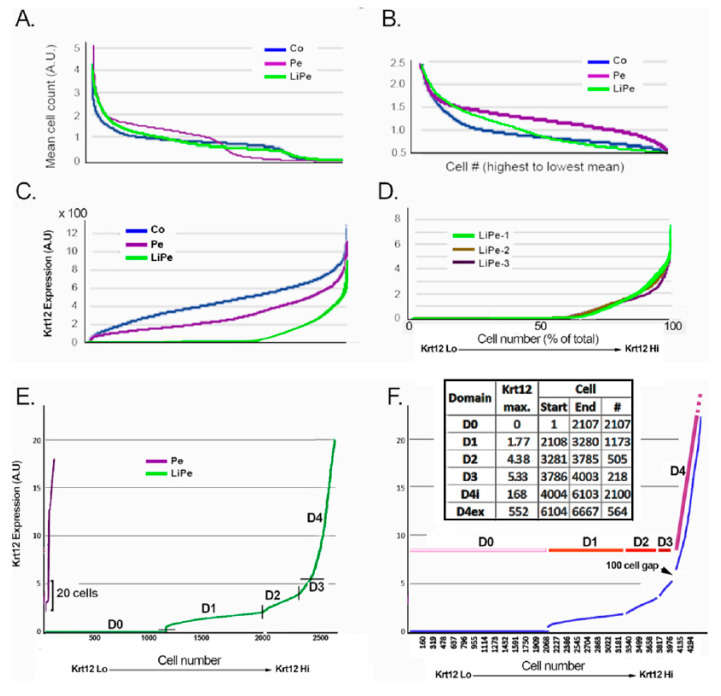
Mean gene count distribution and relative Krt12 expression in the limbal–corneal epithelial samples. (**A**) Mean gene count distribution of the cell population after excluding non-epithelial and conjunctival cells. In all three corneal related samples, at a certain point, the mean count undergoes a rapid decrease. At the high end of the count, there are small populations of high cell count. (**B**) The mean gene count distribution following exclusion of the very high and low gene count populations. (**C**) Krt12 gene expression distribution in the LiPe, Pe, and Co populations. Only the LiPe contains cells with nil or extremely low Krt12 gene expression. (**D**) The distribution of Krt12 gene expression in three independent scRNA-Seq experiments. (**E**) The distribution of Krt12 gene expression within the 0–20 A.U. range in the LiPe-1 and Pe-1 samples. Inflections in the rate of increase in Krt12 expression point to five distinct domains, D0 through D4. In the Pe sample, there are no Krt12 nil cells, and only 20 cells have expression levels matching the expression range for the D0–D3 domain. (**F**) The distribution of Krt12 gene expression within the 0–20 A.U. range in the amalgamated LiPe1-LiPe2-LiPe3 sample. Note the near identity of expression ranges with the LiPe-1 shown in panel E. Gaps between domains represent the exclusion of 20 cells at the domain interfaces in the comparative domain analyses. In the table, D4 has been split into two subdomains, D4i and D4ex.

**Figure 3 biology-13-00145-f003:**
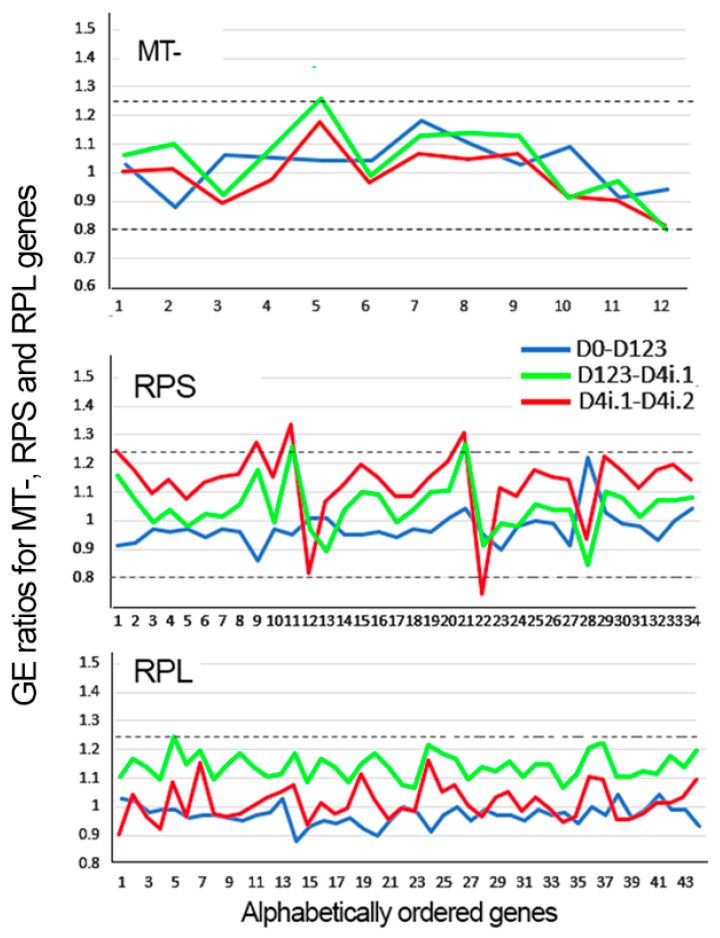
Ratios for each gene included in the MT-, RPS, and RPL gene sets. The dotted lines indicate the 1.25x ratios in either direction.

**Figure 4 biology-13-00145-f004:**
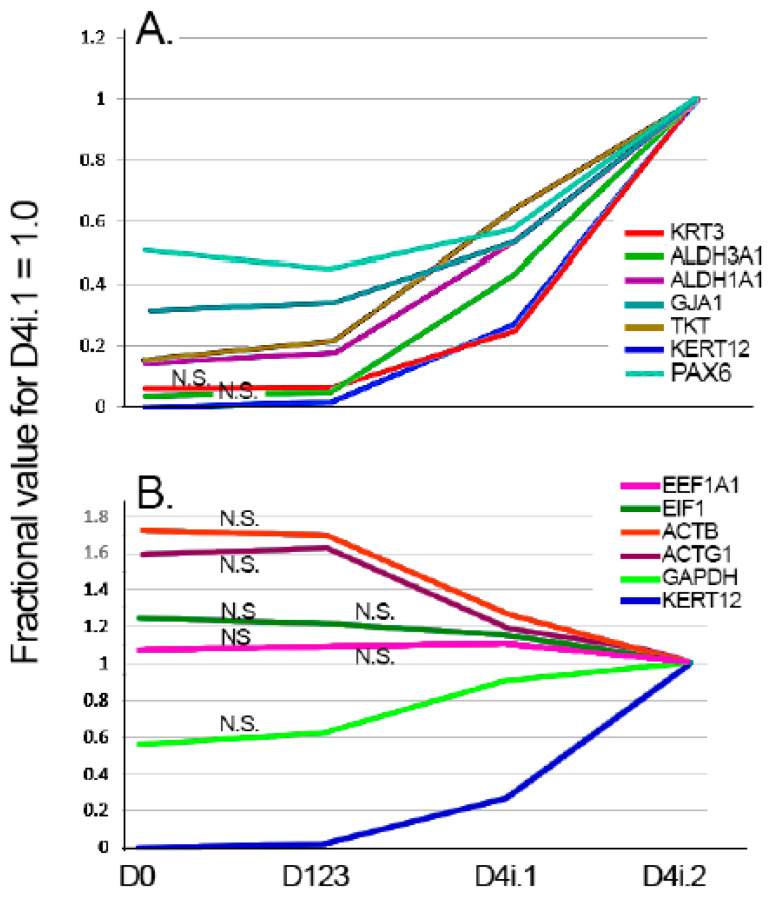
Expression relationships between Krt12 and selected high expression genes with expected behavior. (**A**) Genes expected to increase with differentiation. (**B**) Genes expected to be invariant across the spectrum of Krt12 increase. N.S. Not statistically significant.

**Figure 5 biology-13-00145-f005:**
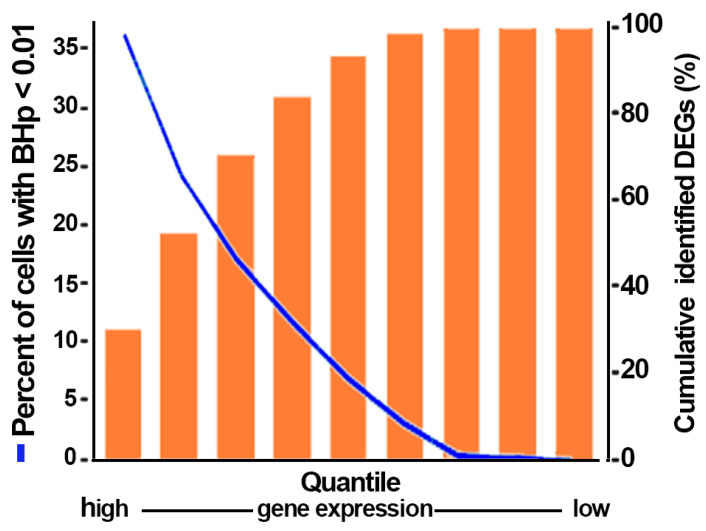
The effect of cell expression level on the number of genes complying with the BH-*p* < 0.01 threshold. Each quantile (Q) consists of 2000 genes in decreasing overall gene expression for a total of 18,000 examined genes.

**Figure 6 biology-13-00145-f006:**
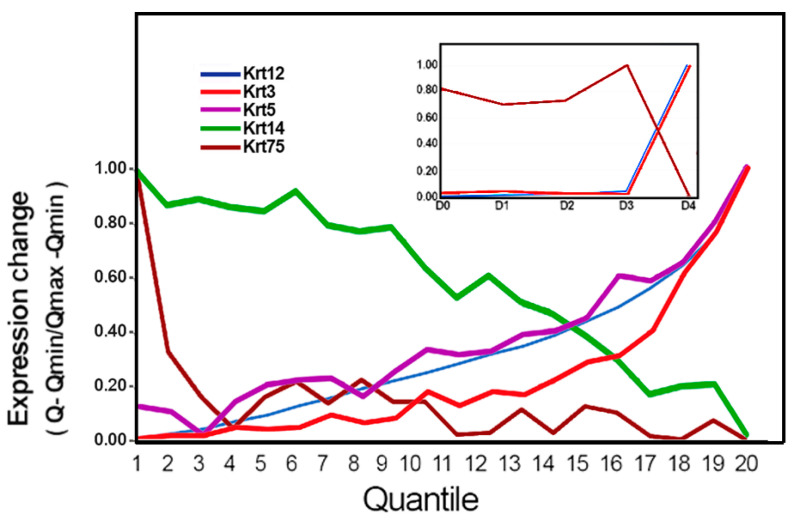
Correlation of GE changes with the changes in Krt12 for cytokeratins.

**Figure 7 biology-13-00145-f007:**
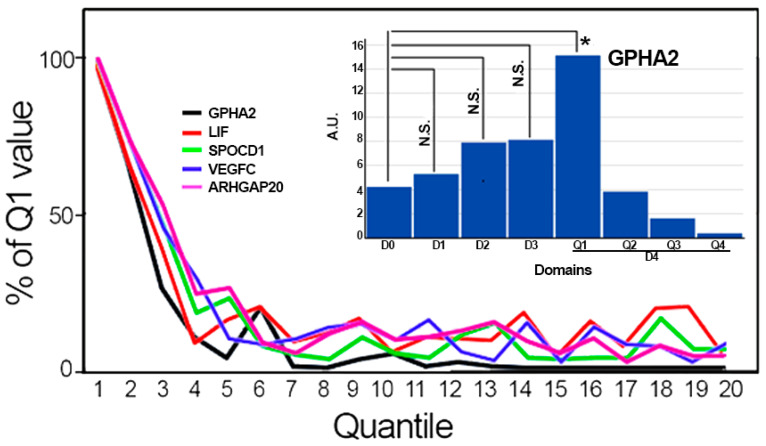
Main frame: The distribution of GPHA2 and four highly correlated genes in the first 2000 cells of the Pe domain of LiPe. Inset: The distribution of GPHA2 between the D0–D3 domains and D4i split into 4 quantiles. N.S., not significant; * BHp = 0.0003.

**Table 1 biology-13-00145-t001:** Accounting of cell number and calculated cell types present in each scRNA seq sample. * as a percent of epithelial.

	LiPe-1		LiPe-2		LiPe-3		LiPe Aver.	Periphery-1		Central Cornea-1	
	#	%	#	%	#	%	Average	#	%	#	%
Total cells	5843	100	2234	100	1981	100	100.00	3731	100	7194	100
Melanocytes	585	10.01	143	6.4	40	2	6.14	25	0.67	25	0.35
Langerhans	3	0.05	0	0	1	0.05	0.03	0	0	2	0.03
Blood cells	3	0.05	2	0.09	0	0	0.05	0	0	0	0
CNJE (AQP5)	9	0.15	9	0.4	2	1	0.52	0	0	0	0
Nonpithelial	600	10.3	154	6.9	14	0.7	5.97	25	0.7	27	0.4
Epithelial	5243	89.7	2080	93.1	1924	97.1	93.30	3706	99.3	7167	99.6
0.5–2.5 limit *	3640	69.4	1746	83.09	1281	66.5	72.32	1982	50.98	5060	71.39
Krt12 = 0 *	1118	30.2	543.00	31.3	446.00	34.6	32.00	0	0	0	0

**Table 2 biology-13-00145-t002:** Mean ± SD global ratios between different Krt12 expression domains for all protein coding. Mitochondria (MT-) and small (RPS) and large (RPL) ribosomes. The number of genes in each cohort is given in parenthesis. The *p* values were calculated by comparing the ratios for all genes in between domain pairs (<>).

		D0vD123	<>	D123vD4i.1	<>	D4i.1vD4i.2	D0vD4i.2
MT-n (12)	Mean	1.029		1.052		0.982	1.069
	SD	0.08		0.122		0.092	0.259
	*p*=		0.614		0.143		
RPS (34)	Mean	0.973		1.031		1.128	1.126
	SD	0.058		0.102		0.114	0.253
	*p*=		0.164		0.131		
RPL (44)	Mean	0.971		1.019		1.128	1.124
	SD	0.034		0.06		0.043	0.101
	*p*=		1.90 × 10^−5^		1.42 × 10^−16^		

**Table 3 biology-13-00145-t003:** Accounting of down and upregulated DEGs within the domains (D) of the LiPe sample and the start and end Quantiles of the Co sample. The total number of DEGs and the number exceeding a 1.35-fold change for the 6647 genes compared are shown.

DEG	D0–D1		D1–D23		D0–D123		D123–Di4.1		Di (4.1–4.2)		Co Q^start^/Q^end^	
	down	up	down	up	down	up	down	up	down	up	down	up
All	617	409	0	16	885	526	2383	1622	1254	1129	2002	4277
R > 1.35x	137		0		160		1456		518		1393	
1/R > 1.35x		79		8		133		1240		515		3380

**Table 4 biology-13-00145-t004:** Top genes that are downregulated (top) or upregulated (bottom) with the initiation of Krt12 expression (D0 to D1) within the limbal domain.

#	Gene	BHp	R	#	Gene	BHp=	R	#	Gene	BHp	R
** *1* **	GRASP	8.55 × 10^−7^	1.76	** *21* **	HSPA2	3.84 × 10^−7^	1.55	** *41* **	TSPAN13	3.13 × 10^−6^	1.47
** *2* **	NR2F2	3.27 × 10^−6^	1.75	** *22* **	USP31	1.40 × 10^−6^	1.55	** *42* **	PHLPP1	1.66 × 10^−5^	1.47
** *3* **	ARC	9.04 × 10^−4^	1.71	** *23* **	SLC2A4RG	1.32 × 10^−4^	1.54	** *43* **	FZD7	2.62 × 10^−4^	1.47
** *4* **	IER5L	3.08 × 10^−4^	1.7	** *24* **	ABHD17B	1.47 × 10^−4^	1.54	** *44* **	CFD	9.02 × 10^−4^	1.47
** *5* **	ANKRD37	4.70 × 10^−4^	1.67	** *25* **	AC027031.2	9.36 × 10^−4^	1.54	** *45* **	EMG1	2.97 × 10^−5^	1.46
** *6* **	OSGIN1	1.05 × 10^−6^	1.66	** *26* **	STMN1	5.29 × 10^−9^	1.53	** *46* **	EID2	3.20 × 10^−4^	1.46
** *7* **	NPPC	6.20 × 10^−5^	1.66	** *27* **	THAP11	3.84 × 10^−4^	1.53	** *47* **	AL035071.1	1.30 × 10^−3^	1.46
** *8* **	SOCS1	1.11 × 10^−5^	1.65	** *28* **	EIF1AY	4.13 × 10^−5^	1.51	** *48* **	GYG1	2.56 × 10^−3^	1.46
** *9* **	POU3F1	3.88 × 10^−8^	1.64	** *29* **	FOS	1.32 × 10^−4^	1.51	** *49* **	TSLP	4.22 × 10^−3^	1.46
** *10* **	NCKAP5L	3.86 × 10^−7^	1.63	** *30* **	CEBPA	1.14 × 10^−3^	1.51	** *50* **	CEP290	5.15 × 10^−3^	1.46
** *11* **	TUSC1	7.90 × 10^−5^	1.62	** *31* **	CBX4	3.07 × 10^−6^	1.5	** *51* **	MYLIP	7.66 × 10^−7^	1.45
** *12* **	cyp26	2.07 × 10^−3^	1.61	** *32* **	TXNDC11	3.09 × 10^−4^	1.5	** *52* **	PER1	8.34 × 10^−6^	1.45
** *13* **	TNFSF9	2.70 × 10^−5^	1.6	** *33* **	BDKRB1	4.57 × 10^−3^	1.5	** *53* **	EGLN1	8.49 × 10^−4^	1.45
** *14* **	TPM2	5.71 × 10^−4^	1.6	** *34* **	JUND	9.95 × 10^−16^	1.49	** *54* **	NFIB	4.14 × 10^−5^	1.44
** *15* **	SCARA3	1.28 × 10^−3^	1.6	** *35* **	HMGB2	4.86 × 10^−7^	1.49	** *55* **	RHOB	2.80 × 10^−4^	1.44
** *16* **	TMEM263	1.03 × 10^−8^	1.59	** *36* **	POMZP3	2.93 × 10^−5^	1.49	** *56* **	ALKBH5	8.78 × 10^−4^	1.44
** *17* **	ID3	1.85 × 10^−3^	1.59	** *37* **	TMEM138	8.32 × 10^−5^	1.49	** *57* **	VEGFC	2.85 × 10^−3^	1.44
** *18* **	ARL4A	1.38 × 10^−7^	1.57	** *38* **	MEIS2	8.26 × 10^−5^	1.48	** *58* **	CXCL8	3.42 × 10^−3^	1.44
** *19* **	GEM	3.59 × 10^−4^	1.57	** *39* **	HINT3	4.79 × 10^−4^	1.48	** *59* **	EGOT	3.90 × 10^−3^	1.44
** *20* **	H2AFX	2.50 × 10^−11^	1.56	** *40* **	IRX2	2.64 × 10^−10^	1.47	** *60* **	ID4	7.11 × 10^−7^	1.43
	**Gene**	**BHp**	**1/R**		**Gene**	**BHp**	**1/R**		**Gene**	**BHp**	**1/R**
** *1* **	SPRR2A	1.06 × 10^−5^	2.50	** *21* **	AC010503.4	3.38 × 10^−4^	1.54	** *41* **	FAM83A	5.23 × 10^−4^	1.43
** *2* **	LY6D	8.83 × 10^−4^	2.44	** *22* **	ABAT	2.43 × 10^−3^	1.54	** *42* **	GALNT7	9.78 × 10^−3^	1.43
** *3* **	SCGB2A1	1.37 × 10^−4^	2.22	** *23* **	ABCA12	5.66 × 10^−3^	1.54	** *43* **	RHOD	4.30 × 10^−8^	1.41
** *4* **	UPK1B	3.01 × 10^−5^	2.00	** *24* **	TRIP10	1.00 × 10^−6^	1.52	** *44* **	S100A14	2.14 × 10^−6^	1.41
** *5* **	MGST1	4.08 × 10^−4^	1.89	** *25* **	POLR2J3	4.33 × 10^−5^	1.52	** *45* **	SULT2B1	2.98 × 10^−6^	1.41
** *6* **	SH3KBP1	3.06 × 10^−4^	1.85	** *26* **	GPX2	1.28 × 10^−3^	1.52	** *46* **	PTPRE	1.30 × 10^−5^	1.41
** *7* **	KRT13	2.35 × 10^−5^	1.82	** *27* **	CLEC7A	4.47 × 10^−3^	1.52	** *47* **	ALDH3A1	9.06 × 10^−5^	1.41
** *8* **	CLCA2	5.82 × 10^−5^	1.75	** *28* **	CRTAC1	2.99 × 10^−6^	1.49	** *48* **	ZNF836	9.10 × 10^−5^	1.41
** *9* **	MFSD4A	2.53 × 10^−3^	1.75	** *29* **	TJP2	5.08 × 10^−6^	1.49	** *49* **	AHNAK2	1.01 × 10^−4^	1.41
** *10* **	CDKN2A	4.09 × 10^−3^	1.75	** *30* **	PLAT	3.42 × 10^−3^	1.49	** *50* **	RAB11FIP1	4.20 × 10^−4^	1.41
** *11* **	CXCL17	7.30 × 10^−3^	1.72	** *31* **	CD24	6.01 × 10^−3^	1.49	** *51* **	CEACAM1	1.34 × 10^−3^	1.41
** *12* **	KRT6A	6.04 × 10^−11^	1.69	** *32* **	APOBEC3A	3.23 × 10^−7^	1.47	** *52* **	ZNF407	2.93 × 10^−3^	1.41
** *13* **	DSG1	2.74 × 10^−10^	1.67	** *33* **	LINC02178	6.87 × 10^−3^	1.47	** *53* **	MYO5B	3.01 × 10^−9^	1.39
** *14* **	NME2	7.30 × 10^−9^	1.61	** *34* **	TMEM238	3.18 × 10^−5^	1.45	** *54* **	NQO1	4.15 × 10^−7^	1.39
** *15* **	LINC02154	4.51 × 10^−4^	1.61	** *35* **	TRPV3	1.26 × 10^−3^	1.45	** *55* **	ADIRF	7.09 × 10^−7^	1.39
** *16* **	S100A4	3.14 × 10^−8^	1.59	** *36* **	EMP1	2.67 × 10^−15^	1.43	** *56* **	TNFRSF10A	1.62 × 10^−6^	1.39
** *17* **	SLPI	8.93 × 10^−4^	1.59	** *37* **	CST3	2.75 × 10^−9^	1.43	** *57* **	IL1RN	2.42 × 10^−6^	1.39
** *18* **	LYPD3	5.15 × 10^−9^	1.56	** *38* **	SDCBP2	7.61 × 10^−8^	1.43	** *58* **	FRMD8	1.45 × 10^−4^	1.39
** *19* **	CDKN2B	3.04 × 10^−6^	1.54	** *39* **	GRHL1	2.69 × 10^−5^	1.43	** *59* **	AC016831.1	1.14 × 10^−3^	1.39
** *20* **	PTGES	9.10 × 10^−5^	1.54	** *40* **	TMPRSS4	3.91 × 10^−4^	1.43	** *60* **	PTPN21	1.90 × 10^−3^	1.39

**Table 5 biology-13-00145-t005:** Top downregulated (top) or upregulated (bottom) genes in the transition from the Krt12-positive limbal cells (D123) to the first 1000 cells of the D4 (i.e., Pe) domain.

#	Gene	BHp	R	#	Gene	Bhp	R	#	Gene	BHp	R
** *1* **	DOCK10	3.45 × 10^−18^	6.86	** *21* **	IL1RL2	1.56 × 10^−19^	4.06	**41**	TBX3	5.32 × 10^−10^	**3.41**
** *2* **	TAGLN	2.26 × 10^−6^	6.23	** *22* **	TPM2	7.72 × 10^−13^	3.97	**42**	WFDC2	3.56 × 10^−29^	**3.40**
** *3* **	PAPPA	9.99 × 10^−16^	5.85	** *23* **	IL23A	1.86 × 10^−18^	3.94	**43**	C6orf141	1.05 × 10^−54^	3.40
** *4* **	GBP2	4.70 × 10^−13^	5.50	** *24* **	FGF13	5.11 × 10^−27^	3.90	**44**	S100A2	6.0 × 10^−137^	3.39
** *5* **	KCNH8	2.76 × 10^−16^	5.47	** *25* **	PTGER4	6.07 × 10^−32^	3.90	**45**	TAC1	2.49 × 10^−6^	3.38
** *6* **	C2CD4A	1.45 × 10^−6^	5.32	** *26* **	AC145124	1.75 × 10^−29^	3.83	**46**	TMEM158	1.11 × 10^−49^	3.34
** *7* **	LPAR6	3.46 × 10^−6^	5.11	** *27* **	TNC	2.07 × 10^−20^	3.81	**47**	SERPINE2	1.29 × 10^−12^	3.32
** *8* **	GSDME	1.20 × 10^−24^	4.99	** *28* **	C12orf54	7.82 × 10^−17^	3.79	**48**	RCAN1	7.53 × 10^−19^	3.32
** *9* **	MSN	2.0 × 10^−105^	4.99	** *29* **	ACTN1	3.74 × 10^−98^	3.74	**49**	PDE4B	1.89 × 10^−41^	3.29
** *10* **	LIF	4.37 × 10^−17^	4.84	** *30* **	FERMT2	5.04 × 10^−62^	3.73	**50**	MMP10	2.12 × 10^−11^	3.27
** *11* **	VEGFC	1.70 × 10^−12^	4.59	** *31* **	TRAC	2.09 × 10^−34^	3.71	**51**	TMEM173	3.73 × 10^−15^	3.26
** *12* **	TCEAL2	4.70 × 10^−13^	4.55	** *32* **	ZNF711	1.54 × 10^−12^	3.66	**52**	ARHGAP20	1.43 × 10^−17^	3.26
** *13* **	SERPINE1	2.81 × 10^−29^	4.48	** *33* **	CDH3	9.05 × 10^−92^	3.64	**53**	GPAT3	8.79 × 10^−23^	3.24
** *14* **	EGR4	2.15 × 10^−8^	4.47	** *34* **	EMP3	4.83 × 10^−64^	3.56	**54**	CALD1	2.05 × 10^−34^	3.20
** *15* **	SPOCD1	4.61 × 10^−25^	4.42	** *35* **	SERPING1	5.16 × 10^−21^	3.56	**55**	FXYD5	3.66 × 10^−32^	3.18
** *16* **	SLC43A3	1.29 × 10^−20^	4.42	** *36* **	C1R	2.49 × 10^−22^	3.52	**56**	LIMA1	9.22 × 10^−89^	3.17
** *17* **	BEX1	1.17 × 10^−8^	4.37	** *37* **	SPINK6	2.92 × 10^−13^	3.49	**57**	VCL	7.73 × 10^−99^	3.14
** *18* **	SERPINB10	1.45 × 10^−36^	4.26	** *38* **	NPPC	5.06 × 10^−13^	3.46	**58**	DIRAS3	6.40 × 10^−13^	3.14
** *19* **	PPP1R12B	4.14 × 10^−13^	4.18	** *39* **	TRAF1	3.22 × 10^−11^	3.44	**59**	HLA-G	2.57 × 10^−14^	3.12
** *20* **	IL1R2	4.77 × 10^−16^	4.13	** *40* **	ADORA2B	2.26 × 10^−19^	3.43	**60**	MUC1	2.09 × 10^−18^	3.12
** *#* **	**Gene**	**Bhp**	**1/R**	**#**	**Gene**	**Bhp**	**1/R**	**#**	**Gene**	**Bhp**	**1/R**
** *1* **	LINC01474	1.92 × 10^−33^	14.43	** *21* **	GCHFR	5.79 × 10^−49^	4.15	**41**	ADIRF	2.4 × 10^−139^	3.30
** *2* **	ALDH3A1	3.70 × 10^−156^	9.49	** *22* **	FSIP1	6.59 × 10^−40^	4.07	**42**	NTRK2	5.89 × 10^−29^	3.28
** *3* **	KRTAP4-1	1.19 × 10^−7^	9.20	** *23* **	SAMD9	2.98 × 10^−25^	3.97	**43**	CPXM2	1.74 × 10^−64^	3.28
** *4* **	KRT3	1.11 × 10^−20^	8.64	** *24* **	ERICH5	3.26 × 10^−70^	3.94	**44**	WNT4	2.47 × 10^−47^	3.28
** *5* **	CRTAC1	1.61 × 10^−304^	8.31	** *25* **	AGR2	6.20 × 10^−48^	3.88	**45**	ABAT	4.69 × 10^−42^	3.23
** *6* **	HTRA1	6.30 × 10^−65^	8.08	** *26* **	CAPS	6.57 × 10^−35^	3.83	**46**	RAB40B	3.61 × 10^−53^	3.20
** *7* **	SPOCK1	1.28 × 10^−37^	7.89	** *27* **	AC025164.1	5.76 × 10^−28^	3.80	**47**	NIPAL3	1.18 × 10^−27^	3.19
** *8* **	MAL	3.88 × 10^−24^	7.05	** *28* **	EPB41L4B	8.75 × 10^−42^	3.71	**48**	AMPD3	5.75 × 10^−33^	3.19
** *9* **	UPK1B	7.79 × 10^−169^	6.84	** *29* **	ALDH1A1	3.1 × 10^−223^	3.70	**49**	SPINK1	3.72 × 10^−5^	3.18
** *10* **	MISP	4.20 × 10^−38^	6.64	** *30* **	GSTA4	3.71 × 10^−46^	3.69	**50**	PNKD	2.45 × 10^−67^	3.15
** *11* **	CLCA2	2.38 × 10^−117^	5.73	** *31* **	MIR193BHG	6.06 × 10^−37^	3.69	**51**	GLRX	3.67 × 10^−7^	3.14
** *12* **	HRK	5.55 × 10^−40^	5.61	** *32* **	TSPAN1	2.00 × 10^−57^	3.64	**52**	LINC01705	6.23 × 10^−10^	3.13
** *13* **	TGFBI	1.10 × 10^−217^	5.09	** *33* **	S100A4	8.4 × 10^−194^	3.62	**53**	CDKN2A	3.06 × 10^−28^	3.12
** *14* **	CALML3	2.88 × 10^−29^	4.96	** *34* **	PPP1R3C	6.73 × 10^−18^	3.60	**54**	SCGB2A1	3.90 × 10^−23^	3.05
** *15* **	SPINK7	5.12 × 10^−11^	4.92	** *35* **	TKT	5.2 × 10^−299^	3.54	**55**	TRPV3	2.98 × 10^−47^	3.05
** *16* **	FA2H	1.07 × 10^−78^	4.52	** *36* **	KRTAP4-6	3.19 × 10^−9^	3.50	**56**	PTGS2	6.68 × 10^−43^	3.05
** *17* **	NQO1	2.51 × 10^−156^	4.50	** *37* **	MFSD4A	9.16 × 10^−33^	3.39	**57**	THBD	9.75 × 10^−13^	3.03
** *18* **	DAPL1	6.94 × 10^−210^	4.43	** *38* **	LINC00707	1.57 × 10^−29^	3.37	**58**	KRT24	4.76 × 10^−4^	3.02
** *19* **	FABP4	2.92 × 10^−5^	4.33	** *39* **	KRTAP3-2	1.52 × 10^−9^	3.37	**59**	SCARA3	2.50 × 10^−27^	3.00
** *20* **	PIR	1.25 × 10^−56^	4.29	** *40* **	SLAMF7	5.72 × 10^−21^	3.32	**60**	GAMT	1.56 × 10^−52^	2.97

**Table 6 biology-13-00145-t006:** The top downregulated and upregulated genes in the comparison of the D4i, first and second 1000 cell subdomains.

#	Gene	BHp	R	#	Gene	BHp	R	#	Gene	BHp	R
** *1* **	GPHA2	3.54 × 10^−6^	26.30	**21**	SERPINE2	1.81 × 10^−6^	3.96	**41**	CXCL8	5.10 × 10^−4^	2.88
** *2* **	CLEC2D	1.44 × 10^−3^	11.85	**22**	WFDC2	3.01 × 10^−11^	3.62	**42**	C2orf54	1.60 × 10^−4^	2.80
** *3* **	C2CD4A	3.58 × 10^−3^	7.53	**23**	DIRAS3	1.01 × 10^−6^	3.53	**43**	DOCK10	9.15 × 10^−4^	2.75
** *4* **	CXCL1	8.61 × 10^−6^	6.60	**24**	KRT75	2.27 × 10^−3^	3.53	**44**	FGF2	2.63 × 10^−7^	2.75
** *5* **	CA2	1.72 × 10^−4^	5.68	**25**	C2CD4B	5.64 × 10^−3^	3.44	**45**	TSLP	2.61 × 10^−7^	2.72
** *6* **	SPOCD1	5.05 × 10^−8^	5.52	**26**	SERPINB10	7.67 × 10^−6^	3.29	**46**	SERPING1	1.31 × 10^−5^	2.71
** *7* **	TCEAL2	9.95 × 10^−5^	5.44	**27**	CH25H	6.73 × 10^−5^	3.28	**47**	CCL20	7.46 × 10^−4^	2.68
** *8* **	PLTP	4.92 × 10^−19^	5.41	**28**	PTGER4	2.97 × 10^−9^	3.19	**48**	TNC	8.99 × 10^−7^	2.61
** *9* **	PAPPA	1.26 × 10^−4^	5.36	**29**	F3	4.79 × 10^−7^	3.14	**49**	PDE4B	2.39 × 10^−13^	2.61
** *10* **	TRAC	5.54 × 10^−13^	5.11	**30**	NPPC	4.46 × 10^−6^	3.12	**50**	S100A3	5.73 × 10^−14^	2.60
** *11* **	LUM	5.31 × 10^−4^	4.98	**31**	TPM2	1.12 × 10^−3^	3.09	**51**	FXYD5	5.98 × 10^−11^	2.60
** *12* **	LIF	6.72 × 10^−5^	4.81	**32**	GLIPR1	4.09 × 10^−5^	3.08	**52**	APOE	1.83 × 10^−10^	2.58
** *13* **	KCNH8	1.32 × 10^−4^	4.60	**33**	MSN	1.47 × 10^−18^	3.06	**53**	C1R	1.06 × 10^−6^	2.54
** *14* **	SPINK6	1.81 × 10^−6^	4.52	**34**	GULP1	8.50 × 10^−10^	3.05	**54**	TBX3	6.71 × 10^−4^	2.51
** *15* **	C12orf54	1.59 × 10^−5^	4.46	**35**	INHBA	3.29 × 10^−4^	3.00	**55**	NTNG2	2.55 × 10^−4^	2.50
** *16* **	ARHGAP20	1.84 × 10^−6^	4.45	**36**	TMEM158	9.55 × 10^−17^	2.98	**56**	CCDC88A	7.76 × 10^−9^	2.50
** *17* **	VEGFC	5.06 × 10^−5^	4.29	**37**	FGFR1	3.44 × 10^−14^	2.95	**57**	SOD2	4.35 × 10^−7^	2.47
** *18* **	GBP2	1.64 × 10^−4^	4.27	**38**	GBP1	4.16 × 10^−7^	2.95	**58**	MT-ND6	4.22 × 10^−12^	2.46
** *19* **	BEX1	5.22 × 10^−3^	4.21	**39**	S100A2	2.74 × 10^−38^	2.94	**59**	H1F0	2.06 × 10^−4^	2.46
** *20* **	SLC43A3	7.12 × 10^−5^	4.09	**40**	PXDN	4.36 × 10^−6^	2.91	**60**	CALD1	5.24 × 10^−9^	2.44
**#**	**Gene**	**BHp**	**1/R**	**#**	**Gene**	**BHp**	**1/R**	**#**	**Gene**	**BHp**	**1/R**
** *1* **	HSPA6	5.53 × 10^−5^	5.03	** *21* **	FA2H	4.49 × 10^−33^	2.07	** *41* **	ANKRD37	3.77 × 10^−8^	1.88
** *2* **	KRT3	1.74 × 10^−16^	4.85	** *22* **	HSPA1B	1.39 × 10^−12^	2.06	** *42* **	GJA1	2.78 × 10^−45^	1.88
** *3* **	LINC01474	1.08 × 10^−28^	3.67	** *23* **	SCARA3	2.03 × 10^−17^	2.06	** *43* **	SMIM30	6.04 × 10^−16^	1.88
** *4* **	KRTAP4-6	2.31 × 10^−7^	3.34	** *24* **	MRPL33	1.25 × 10^−79^	2.06	** *44* **	HOMER3	3.61 × 10^−28^	1.88
** *5* **	CALML3	6.58 × 10^−21^	2.89	** *25* **	AGR2	1.17 × 10^−21^	2.05	** *45* **	HLA-DRA	2.73 × 10^−6^	1.86
** *6* **	SPOCK1	9.22 × 10^−30^	2.88	** *26* **	PIR	7.67 × 10^−24^	2.05	** *46* **	PSAT1	2.01 × 10^−28^	1.86
** *7* **	ERICH5	5.26 × 10^−51^	2.75	** *27* **	MISP	5.14 × 10^−13^	1.99	** *47* **	MUC15	1.44 × 10^−12^	1.86
** *8* **	HTRA1	6.11 × 10^−21^	2.46	** *28* **	SCIN	2.62 × 10^−18^	1.96	** *48* **	RAD9A	1.72 × 10^−11^	1.85
** *9* **	MAL	1.13 × 10^−13^	2.38	** *29* **	CXCL14	9.58 × 10^−25^	1.95	** *49* **	PNKD	3.24 × 10^−32^	1.84
** *10* **	GYG1	2.56 × 10^−25^	2.31	** *30* **	SNCG	1.75 × 10^−12^	1.95	** *50* **	PDHB	2.90 × 10^−14^	1.84
** *11* **	MGARP	2.83 × 10^−55^	2.24	** *31* **	BEX3	1.26 × 10^−29^	1.92	** *51* **	WNT4	3.89 × 10^−18^	1.83
** *12* **	TSPAN1	1.35 × 10^−40^	2.22	** *32* **	DNAJB4	1.73 × 10^−9^	1.91	** *52* **	RRAD	2.50 × 10^−5^	1.83
** *13* **	HSPA1A	4.06 × 10^−14^	2.19	** *33* **	DGCR6L	2.44 × 10^−20^	1.91	** *53* **	METTL5	1.44 × 10^−18^	1.82
** *14* **	ALDH3A1	1.22 × 10^−46^	2.17	** *34* **	SERINC2	1.56 × 10^−25^	1.90	** *54* **	NEDD9	6.98 × 10^−18^	1.82
** *15* **	CAPNS2	6.60 × 10^−17^	2.16	** *35* **	NQO1	8.90 × 10^−39^	1.89	** *55* **	RASSF6	2.20 × 10^−16^	1.81
** *16* **	TP53I3	1.14 × 10^−32^	2.15	** *36* **	DAPL1	6.04 × 10^−61^	1.89	** *56* **	EPS8L2	2.12 × 10^−15^	1.81
** *17* **	CLCA4	4.86 × 10^−13^	2.15	** *37* **	ALDH1A1	2.83 × 10^−95^	1.89	** *57* **	PAX6	1.72 × 10^−20^	1.81
** *18* **	GJB6	2.79 × 10^−56^	2.14	** *38* **	ECM1	2.61 × 10^−19^	1.88	** *58* **	CRTAC1	1.58 × 10^−71^	1.81
** *19* **	KIF22	6.36 × 10^−16^	2.13	** *39* **	FSIP1	1.30 × 10^−13^	1.88	** *59* **	FAM114A1	5.06 × 10^−17^	1.81
** *20* **	CAPS	2.97 × 10^−19^	2.10	** *40* **	OTUD1	2.21 × 10^−7^	1.88	** *60* **	LAMTOR2	6.57 × 10^−29^	1.80

**Table 7 biology-13-00145-t007:** Upper panels: Top genes that undergo continuous down or (left top panel) upregulation (right top panel) along the whole Krt12 expression span in the LiPe sample. R is the D0/D4i.2 expression ratio; 1/R is the D4i.2/D0 ratio. Lower panels: Top genes that undergo down (bottom left panel) or upregulation (bottom right panel) only at the D0 to D1 transition. R1 and R123 are the D0/D1 and D0/D123 expression ratios, respectively; 1/R1 and 1/R123 are the D0/D1 (R1) and D0/D123 R123 ratios, respectively. The genes with ratios below 1.35 are displayed in italics.

#	Gene	R	#	Gene	R	#	Gene	1/R	*#*	Gene	1/R
** *1* **	VEGFC	25.37	** *21* **	HSPB8	4.68	** *1* **	ALDH3A1	28.92	** *21* **	ECM1	4.50
** *2* **	TRAC	22.44	** *22* **	SLC12A2	4.57	** *2* **	UPK1B	27.18	** *22* **	DSG1	4.48
** *3* **	TPM2	20.38	** *23* **	SLC6A6	4.52	** *3* **	CRTAC1	26.76	** *23* **	CLU	4.45
** *4* **	NPPC	17.20	** *24* **	TSC22D1	4.37	** *4* **	CLCA2	18.94	** *24* **	ENO1	4.42
** *5* **	CXCL8	10.73	** *25* **	CCDC66	4.26	** *5* **	NQO1	12.59	** *25* **	POLR2J3	4.39
** *6* **	ID4	8.15	** *26* **	TRIM27	4.23	** *6* **	MFSD4A	9.10	** *26* **	TRIP10	4.29
** *7* **	SAPCD2	7.97	** *27* **	LARP6	4.14	** *7* **	ALDH1A1	9.00	** *27* **	CTSD	4.26
** *8* **	cyp26	7.52	** *28* **	MTERF3	3.92	** *8* **	S100A4	8.94	** *28* **	TMEM238	4.06
** *9* **	ZNF22	7.41	** *29* **	MBD3	3.90	** *9* **	SCGB2A1	8.57	** *29* **	FAM83A	3.81
** *10* **	TSLP	7.01	** *30* **	CDC42EP1	3.89	** *10* **	TKT	7.87	** *30* **	ABCA12	3.78
** *11* **	CREB5	6.46	** *31* **	C12orf65	3.87	** *11* **	ADIRF	7.80	** *31* **	GALNT7	3.65
** *12* **	ATP1B1	5.92	** *32* **	FOXC1	3.85	** *12* **	ABAT	7.13	** *32* **	AC010503.4	3.56
** *13* **	PLEKHO1	5.44	** *33* **	TNFAIP8	3.79	** *13* **	CLEC7A	6.95	** *33* **	ASPH	3.30
** *14* **	SPATA2L	5.28	** *34* **	COQ10A	3.75	** *14* **	PTGES	6.41	** *34* **	LMTK2	3.25
** *15* **	MEIS2	5.23	** *35* **	FAM129A	3.74	** *15* **	TRPV3	5.63	** *35* **	GIPC1	3.20
** *16* **	SOX4	5.08	** *36* **	DDIT3	3.69	** *16* **	TMPRSS4	5.42	** *36* **	ARL5B	3.11
** *17* **	ZC2HC1A	5.04	** *37* **	DST	3.66	** *17* **	GSN	5.25	** *37* **	ANXA11	3.08
** *18* **	NCOA7	4.97	** *38* **	DDX28	3.60	** *18* **	CST3	4.88	** *38* **	EMP1	3.01
** *19* **	PALLD	4.95	** *39* **	MYC	3.59	** *19* **	SDC1	4.76	** *39* **	MYH14	2.83
** *20* **	NUDT11	4.90	** *40* **	Thap7	3.59	** *20* **	GPRC5A	4.73	** *40* **	DBI	2.80
**#**	**Gene**	**R1**	**R123**	** *#* **	**Gene**	**R1**	**R123**	**#**	**Gene**	**1/R1**	**1/R123**
** *1* **	NR2F2	1.75	1.74	** *21* **	SHARPIN	1.41	1.39	** *1* **	*HIST1H1E*	*1.30*	*1.30*
** *2* **	ID3	1.67	1.65	** *22* **	*PRPSAP1*	*1.40*	*1.15*	** *2* **	*ITGA2*	*1.27*	*1.20*
** *3* **	SCARA3	1.60	1.58	** *23* **	*CKS1B*	*1.39*	*1.35*	** *3* **	*SLFN5*	*1.25*	*1.18*
** *4* **	GEM	1.57	1.55	** *24* **	*RFXANK*	*1.38*	*1.23*	** *4* **	*ADAMTS9*	*1.25*	*1.31*
** *5* **	SLC2A4RG	1.54	1.48	** *25* **	*CITED2*	*1.38*	*1.35*	** *5* **	*KIAA1551*	*1.25*	*1.20*
** *6* **	ABHD17B	1.54	1.43	** *26* **	*TOB1*	*1.38*	*1.30*	** *6* **	*GJB5*	*1.23*	*1.19*
** *7* **	THAP11	1.53	1.49	** *27* **	*MAZ*	*1.37*	*1.24*	** *7* **	*SEMA4B*	*1.23*	*1.15*
** *8* **	FOS	1.51	1.36	** *28* **	*TENT5C*	*1.37*	*1.40*	** *8* **	*S100A16*	*1.23*	*1.20*
** *9* **	HMGB2	1.49	1.55	** *29* **	*GLA*	*1.36*	*1.31*	** *9* **	*KRT17*	*1.22*	*1.21*
** *10* **	JUND	1.49	1.43	** *30* **	*KLF4*	*1.36*	*1.28*	** *10* **	*PITPNC1*	*1.22*	*1.13*
** *11* **	TSPAN13	1.47	1.41	** *31* **	*BOLA3*	*1.36*	*1.23*	** *11* **	*SLC38A1*	*1.22*	*1.15*
** *12* **	CEP290	1.46	1.48	** *32* **	*RPS4Y1*	*1.35*	*1.22*	** *12* **	*PLS3*	*1.21*	*1.18*
** *13* **	GYG1	1.46	1.27	** *33* **	*CUEDC2*	*1.35*	*1.23*	** *13* **	*RPS17*	*1.19*	*1.17*
** *14* **	PER1	1.45	1.43	** *34* **	*INAVA*	*1.35*	*1.34*	** *14* **	*AQP3*	*1.19*	*1.14*
** *15* **	EGLN1	1.45	1.32	** *35* **	*MRPS11*	*1.34*	*1.31*	** *15* **	*RPL17*	*1.16*	*1.14*
** *16* **	ALKBH5	1.44	1.30	** *36* **	*ATN1*	*1.33*	*1.26*	** *16* **	*WWTR1*	*1.16*	*1.12*
** *17* **	C9orf3	1.43	1.44	** *37* **	*MXD4*	*1.33*	*1.36*	** *17* **	*GARS*	*1.16*	*1.10*
** *18* **	FBXL15	1.43	1.39	** *38* **	*ATG4D*	*1.33*	*1.30*	** *18* **	*TRIM44*	*1.15*	*1.12*
** *19* **	CLEC2B	1.41	1.40	** *39* **	*VDAC3*	*1.33*	*1.26*	** *19* **	*ZMAT2*	*1.15*	*1.13*
** *20* **	HSPA1A	1.41	1.53	** *40* **	*SUMO3*	*1.31*	*1.21*	**20**	*ISG20*	*1.14*	*1.13*

**Table 8 biology-13-00145-t008:** The top downregulated and upregulated genes associated with the transition from the lowest to the highest Krt12 expression in the central cornea (CoQ^start^-CoQ^end^).

#	Gene	BHp	R	#	Gene	BHp	*R*	#	Gene	BHp	R
**1**	MEG3	4.0 × 10^−17^	76.68	**21**	TAC1	2.8 × 10^−18^	*11.14*	**41**	LINC01127	3.0 × 10^−54^	7.60
**2**	ACTN1	5.2 × 10^−107^	48.81	**22**	KCNMA1	3.3 × 10^−72^	*11.09*	**42**	GPNMB	4.6 × 10^−65^	7.56
**3**	VIT	1.5 × 10^−72^	43.58	**23**	S100A2	1.3 × 10^−65^	*11.05*	**43**	PPIF	2.7 × 10^−189^	7.50
**4**	SPARC	8.3 × 10^−131^	35.52	**24**	CDH3	5.7 × 10^−118^	*10.81*	**44**	SYNJ2	4.9 × 10^−178^	7.31
**5**	MFHAS1	7.1 × 10^−70^	31.24	**25**	NTRK2	2.5 × 10^−71^	*10.77*	**45**	MOXD1	2.1 × 10^−111^	7.30
**6**	CCL20	2.0 × 10^−5^	21.54	**26**	CRABP2	5.9 × 10^−103^	*10.44*	**46**	PALLD	3.8 × 10^−168^	7.18
**7**	DRAM1	2.0 × 10^−45^	18.60	**27**	BMP2	5.6 × 10^−29^	*10.22*	**47**	FST	4.3 × 10^−58^	7.17
**8**	TRAC	3.6 × 10^−36^	18.48	**28**	CAVIN1	8.9 × 10^−191^	*10.01*	**48**	COTL1	3.4 × 10^−47^	6.98
**9**	LGALS1	7.5 × 10^−17^	17.00	**29**	KRT16	2.3 × 10^−43^	*9.78*	**49**	AC020916.1	3.0 × 10^−101^	6.92
**10**	KRT14	3.0 × 10^−264^	16.66	**30**	LAMB1	1.5 × 10^−72^	*9.74*	**50**	MEIS3	5.5 × 10^−85^	6.91
**11**	IL13RA2	2.3 × 10^−46^	16.63	**31**	TMEM158	4.2 × 10^−90^	*9.49*	**51**	OXTR	4.1 × 10^−51^	6.86
**12**	LAMA3	3.4 × 10^−151^	15.95	**32**	NFATC1	1.1 × 10^−58^	*9.31*	**52**	TRIML2	3.7 × 10^−37^	6.78
**13**	TGM2	1.2 × 10^−61^	15.80	**33**	TNNT1	2.3 × 10^−40^	*9.18*	**53**	CPVL	9.1 × 10^−163^	6.77
**14**	ANXA5	0.0 × 10^0^	15.25	**34**	MGLL	3.5 × 10^−37^	*8.53*	**54**	EGR3	7.0 × 10^−42^	6.76
**15**	CCNA1	2.2 × 10^−19^	14.15	**35**	WNT9A	1.3 × 10^−78^	*8.38*	**55**	CDK6	2.3 × 10^−127^	6.75
**16**	SMOX	4.0 × 10^−53^	13.13	**36**	OSBP2	7.1 × 10^−64^	*8.34*	**56**	ZFP42	2.9 × 10^−17^	6.71
**17**	FERMT2	1.4 × 10^−31^	13.07	**37**	FAM180A	6.6 × 10^−27^	*7.96*	**57**	ETS1	1.1 × 10^−145^	6.70
**18**	FLNA	1.9 × 10^−242^	12.68	**38**	SSUH2	4.7 × 10^−46^	*7.85*	**58**	PLAU	1.9 × 10^−223^	6.61
**19**	SALL3	1.2 × 10^−23^	12.30	**39**	SLC7A11	4.4 × 10^−59^	*7.75*	**59**	MSANTD3	1.8 × 10^−52^	6.58
**20**	UGDH	8.3 × 10^−120^	11.79	**40**	SLC9A2	3.8 × 10^−63^	*7.66*	**60**	CARD10	2.1 × 10^−95^	6.58
	**Gene**	**BHp**	**1/R**		**Gene**	**BHp**	** *1/R* **		**Gene**	**BHp**	**1/R**
**1**	CYP26A1	1.3 × 10^−77^	33.40	**21**	SLC26A2	1.4 × 10^−160^	*9.69*	**41**	MUC15	5.3 × 10^−248^	7.88
**2**	PSCA	2.9 × 10^−35^	25.62	**22**	CXXC5	6.3 × 10^−202^	*9.69*	**42**	MUC16	1.4 × 10^−43^	7.80
**3**	APBB1IP	5.4 × 10^−46^	14.64	**23**	RRAD	4.9 × 10^−41^	*9.37*	**43**	SIK1	1.9 × 10^−55^	7.72
**4**	SMIM5	2.9 × 10^−105^	14.25	**24**	TJP3	2.8 × 10^−55^	*9.27*	**44**	FOS	4.9 × 10^−189^	7.70
**5**	CA6	5.8 × 10^−31^	14.06	**25**	DHRS9	1.6 × 10^−15^	*9.20*	**45**	ABLIM2	1.8 × 10^−53^	7.50
**6**	CLIC3	1.3 × 10^−59^	13.92	**26**	HS3ST6	9.9 × 10^−152^	*9.09*	**46**	RAB40C	2.6 × 10^−50^	7.46
**7**	KRTAP4-8	3.9 × 10^−9^	12.87	**27**	TMEM246	4.3 × 10^−46^	*8.95*	**47**	OVOL2	1.9 × 10^−47^	7.31
**8**	ADH7	0.0 × 10^0^	12.12	**28**	RAET1E	6.9 × 10^−96^	*8.94*	**48**	BEND5	1.3 × 10^−60^	7.26
**9**	FAM3D	4.3 × 10^−53^	11.53	**29**	FBP1	7.1 × 10^−112^	*8.75*	**49**	EFS	2.2 × 10^−38^	7.19
**10**	HES5	9.8 × 10^−118^	11.40	**30**	PLBD1	5.5 × 10^−64^	*8.68*	**50**	KRTAP4-9	2.4 × 10^−25^	7.19
**11**	NUDT7	5.3 × 10^−41^	10.96	**31**	ERICH5	0.0 × 10^0^	*8.58*	**51**	RAB6B	7.8 × 10^−159^	6.96
**12**	GNE	3.8 × 10^−48^	10.80	**32**	GGT6	4.8 × 10^−73^	*8.50*	**52**	CXCL2	2.7 × 10^−8^	6.92
**13**	SLURP1	6.9 × 10^−91^	10.70	**33**	RELN	2.3 × 10^−7^	*8.26*	**53**	CAPN5	2.6 × 10^−39^	6.91
**14**	ST6GALNAC1	5.0 × 10^−57^	10.54	**34**	CNGA1	8.2 × 10^−96^	*8.18*	**54**	ICMT	4.7 × 10^−63^	6.87
**15**	ADRB1	1.7 × 10^−44^	10.46	**35**	SCGB2A1	2.2 × 10^−169^	*8.10*	**55**	RGS16	3.1 × 10^−5^	6.83
**16**	CALML5	9.7 × 10^−59^	10.13	**36**	C10orf99	7.0 × 10^−28^	*8.07*	**56**	IL20RA	5.3 × 10^−103^	6.59
**17**	MIR210HG	1.3 × 10^−70^	9.94	**37**	MUC21	4.3 × 10^−20^	*8.06*	**57**	GPX2	1.8 × 10^−157^	6.58
**18**	RHOU	4.5 × 10^−252^	9.93	**38**	TLDC2	9.1 × 10^−51^	*8.05*	**58**	PLEKHH3	2.0 × 10^−118^	6.51
**19**	PCP4L1	7.8 × 10^−66^	9.87	**39**	BBOX1	1.8 × 10^−161^	*7.97*	**59**	HSPA6	1.5 × 10^−10^	6.49
**20**	KRTAP4-7	6.2 × 10^−11^	9.85	**40**	OTUD1	2.7 × 10^−99^	*7.89*	**60**	SLC39A2	4.8 × 10^−30^	6.47

**Table 9 biology-13-00145-t009:** Top frames: Gene ontology terms unique to D0 (left frame) or D1 (right frames) in the D0–D1 comparison set. Bottom frames: Gene ontology terms unique to D123 (left frame) or D4i.1 (right frame) in the D123 to D4i.1 comparison. The fold expression overrepresentation index (FE) and the false discovery rate (FDR) are listed.

D0 Unique Go Terms	FE	FDR	D1 Unique Go Terms	FE	FDR
pos. reg. of core promoter binding	24.65	7.43 × 10^−3^	reg. of plasma membrane organization	15.59	5.12 × 10^−3^
integrated stress response signaling	11.2	2.84 × 10^−5^	cytoplasmic translation	13.34	8.35 × 10^−18^
neg. reg. of mRNA splicing, via spliceosome	9.64	7.73 × 10^−3^	reg. of translation in response to stress	12.75	9.43 × 10^−3^
reg. of transcript. from RNA pol. II promoter in stress	8.45	1.69 × 10^−3^	intermediate filament cytoskeleton organization	6.64	4.93 × 10^−4^
mitochondrial electron transport, NADH to ubiquinone	6.43	6.76 × 10^−3^	ribosomal small subunit biogenesis	5.99	9.17 × 10^−4^
pos. reg. of miRNA metabolic process	6.37	1.38 × 10^−3^	cell–cell junction assembly	5.14	1.35 × 10^−3^
proton motive force-driven mitochondrial ATP synthesis	5.2	9.92 × 10^−3^	keratinocyte differentiation	5.07	7.82 × 10^−4^
aerobic electron transport chain	5.16	1.30 × 10^−3^	epithelial cell development	5.05	5.62 × 10^−5^
neg. reg. of mRNA metabolic process	5.01	8.21 × 10^−4^	actin filament organization	5.01	1.94 × 10^−7^
mitochondrial ATP synthesis coupled electron transport	4.88	1.96 × 10^−3^	pos. reg. of protein-containing complex assembly	3.93	3.61 × 10^−3^
oxidative phosphorylation	4.7	3.58 × 10^−4^	supramolecular fiber organization	3.8	6.86 × 10^−9^
cellular response to topologically incorrect protein	4.57	5.80 × 10^−3^	actin cytoskeleton organization	3.72	6.68 × 10^−8^
reg. of mRNA splicing, via spliceosome	4.53	1.82 × 10^−3^	neg. reg. of apoptotic signaling pathway	3.68	3.85 × 10^−3^
response to unfolded protein	4.35	1.41 × 10^−3^	skin development	3.63	9.20 × 10^−4^
cellular response to leukemia inhibitory factor	4.28	9.10 × 10^−3^	neg. reg. of kinase activity	3.54	8.13 × 10^−3^
reg. of TGFb receptor signaling pathway	3.7	5.40 × 10^−3^	wound healing	3.49	4.16 × 10^−4^
aerobic respiration	3.47	5.76 × 10^−3^	epithelial cell differentiation	3.38	1.95 × 10^−7^
neg. reg. of transcription by RNA polymerase II	3.04	3.36 × 10^−15^	cell junction assembly	3.17	8.53 × 10^−3^
in utero embryonic development	2.95	3.75 × 10^−5^	epidermis development	3.01	6.59 × 10^−3^
reg. of cellular response to growth factor stimulus	2.83	1.40 × 10^−3^	response to wounding	2.95	8.34 × 10^−4^
neg. reg. of DNA-templated transcription	2.82	2.53 × 10^−17^	neg. reg. of catalytic activity	2.94	3.45 × 10^−5^
neg. reg. of RNA biosynthetic process	2.79	4.85 × 10^−17^	neg. reg. of phosphorylation	2.9	9.08 × 10^−3^
neg. reg. of nucleobase-containing metabolic process	2.72	1.72 × 10^−18^	ribonucleoprotein complex biogenesis	2.84	2.00 × 10^−3^
neg. reg. of macromolecule biosynthetic process	2.71	2.03 × 10^−18^	neg. reg. of phosphorus metabolic process	2.77	7.29 × 10^−3^
neg. reg. of cellular biosynthetic process	2.69	1.74 × 10^−18^	pos. reg. of cellular component biogenesis	2.75	1.47 × 10^−3^
reg. of hemopoiesis	2.66	6.53 × 10^−4^	reg. of protein kinase activity	2.65	7.31 × 10^−4^
neg. reg. of biosynthetic process	2.63	4.61 × 10^−18^	reg. of cell migration	2.47	8.07 × 10^−5^
neg. reg. of cellular metabolic process	2.37	1.48 × 10^−18^	cytoskeleton organization	2.38	6.20 × 10^−6^
embryonic organ development	2.34	4.19 × 10^−3^			
**D123 unique GO terms**	**FE**	**FDR**	**D4i.1 unique Go terms**	**FE**	**FDR**
pos. reg. of protein localization to Cajal body	7.01	4.61 × 10^−3^	Golgi lumen acidification	10.14	1.00 × 10^−3^
protein folding in endoplasmic reticulum	7.01	4.58 × 10^−3^	desmosome organization	8.88	4.91 × 10^−3^
maturation of LSU-rRNA	4.9	3.32 × 10^−4^	synaptic vesicle lumen acidification	7.92	4.81 × 10^−4^
maturation of SSU-rRNA	4.78	1.15 × 10^−7^	cellular response to arsenic-containing substance	7.75	2.20 × 10^−4^
maturation of 5.8S rRNA	4.76	3.36 × 10^−5^	lysosomal lumen acidification	5.76	2.87 × 10^−3^
mitochondrial RNA metabolic process	4	1.46 × 10^−4^	vacuolar acidification	4.85	1.19 × 10^−3^
mitochondrial gene expression	3.98	7.60 × 10^−15^	proton motive force-driven mitochondrial ATP synthesis	4.56	4.81 × 10^−6^
neg. reg. of stem cell differentiation	3.98	8.91 × 10^−3^	neg. reg. of stress-activated protein kinase cascade	4.06	9.15 × 10^−4^
spliceosomal snRNP assembly	3.96	7.97 × 10^−4^	mitochondrial ATP synthesis coupled electron transport	3.76	8.48 × 10^−6^
ribosome biogenesis	3.95	1.61 × 10^−32^	ribonucleoside triphosphate metabolic process	3.72	3.02 × 10^−9^
rRNA processing	3.95	1.74 × 10^−22^	cellular aldehyde metabolic process	3.68	8.49 × 10^−4^
pos. reg. of transcription by RNA polymerase I	3.81	2.99 × 10^−3^	glycolytic process	3.66	9.11 × 10^−3^
protein-RNA complex assembly	3.74	3.75 × 10^−18^	NADH dehydrogenase complex assembly	3.48	2.26 × 10^−3^
protein-RNA complex organization	3.72	1.16 × 10^−18^	ERBB signaling pathway	3.17	7.76 × 10^−3^
tRNA aminoacylation for protein translation	3.67	1.61 × 10^−3^	cell–cell junction organization	3.14	1.29 × 10^−7^
reg. of hematop. progenitor cell differentiation	3.35	8.46 × 10^−3^	cellular oxidant detoxification	2.99	2.09 × 10^−3^
reg. of RNA splicing	3.23	2.86 × 10^−7^	pos. reg. of protein localization to membrane	2.93	9.46 × 10^−4^
reg. of telomere maintenance via telomerase	3.18	6.42 × 10^−9^	cellular response to toxic substance	2.9	3.80 × 10^−4^
ncRNA processing	3.17	1.24 × 10^−6^	vacuolar transport	2.7	7.62 × 10^−5^
pos. reg. of protein localization to nucleus	3.1	3.88 × 10^−27^	neg. reg. of apoptotic signaling pathway	2.66	2.32 × 10^−6^
pos. reg. of chromosome organization	3.04	7.98 × 10^−4^	pos. reg. of ubiquitin-dependent prot. catabolic process	2.63	6.03 × 10^−3^
reg. of stem cell population maintenance	2.74	3.95 × 10^−6^	cell–cell junction assembly	2.61	1.81 × 10^−3^
endoplasm. ret.to Golgi vesicle-med. transp.	2.7	6.08 × 10^−4^	organelle disassembly	2.58	9.15 × 10^−3^
reg. of DNA-templated transcription elongation	2.69	6.68 × 10^−24^	process utilizing autophagic mechanism	2.52	1.54 × 10^−7^
peptidyl-lysine acetylation	2.68	6.17 × 10^−3^	energy derivation by oxidation of organic compounds	2.51	4.30 × 10^−6^
reg. of response to endoplasmic ret. stress	2.65	3.18 × 10^−4^	pos. reg. of proteolysis involved in prot. catabolic process	2.48	3.15 × 10^−3^

**Table 10 biology-13-00145-t010:** Top: Genes with a high positive and negative correlation (crl) with the increase in Krt12 expression in the central cornea. Bottom: Genes that correlated with the Krt12 GE increase in D4, Pe, and Co.

#	Gene		Crl	#		Gene	Crl	#	Gene		Crl	#	Gene	Crl		#	Gene		Crl
**1**	SPINT2		0.99	**21**		ERICH5	0.97	**41**	CLU		0.96	**1**	ANXA5	−0.98		**21**	RPS13		−0.95
**2**	KRT5		0.99	**22**		TPD52	0.97	**42**	GJB6		0.96	**2**	FLNA	−0.97		**22**	RPS8		−0.95
**3**	ADIRF		0.99	**23**		NMRK1	0.97	**43**	FXYD3		0.96	**3**	CAVIN1	−0.97		**23**	ILF2		−0.95
**4**	COMT		0.98	**24**		TP53I3	0.97	**44**	PLEKHH3		0.96	**4**	SYNJ2	−0.96		**24**	DEK		−0.95
**5**	SNCG		0.98	**25**		UPK1B	0.97	**45**	WLS		0.96	**5**	EGR3	−0.96		**25**	EFNB2		−0.95
**6**	ASPH		0.98	**26**		SUCO	0.97	**46**	CSRP2		0.96	**6**	CRABP2	−0.96		**26**	CREBRF		−0.95
**7**	MGARP		0.98	**27**		SPOCK1	0.97	**47**	CD99		0.96	**7**	PPIF	−0.96		**27**	H2AFZ		−0.95
**8**	LINC01474		0.98	**28**		PAX6	0.97	**48**	BCAP31		0.96	**8**	PLAU	−0.96		**28**	RPS16		−0.95
**9**	GJA1		0.98	**29**		C4orf3	0.97	**49**	SH3RF1		0.96	**9**	CDK6	−0.96		**29**	RPS23		−0.94
**10**	ASAH1		0.98	**30**		KRT3	0.97	**50**	GLOD4		0.96	**10**	AJAP1	−0.96		**30**	PABPC1		−0.94
**11**	TSTD1		0.98	**31**		TCEA3	0.97	**51**	EFNA1		0.96	**11**	PTMS	−0.95		**31**	RPS19		−0.94
**12**	SCIN		0.98	**32**		PSAT1	0.97	**52**	CRYBG1		0.96	**12**	PALLD	−0.95		**32**	SUB1		−0.94
**13**	SLC20A1		0.98	**33**		PEBP1	0.97	**53**	SLC2A1		0.96	**13**	SLC9A2	−0.95		**33**	MEPCE		−0.94
**14**	GSN		0.98	**34**		ENO1	0.97	**54**	RB1		0.96	**14**	HIVEP1	−0.95		**34**	EIF5		−0.94
**15**	PDP1		0.97	**35**		MFSD4A	0.96	**55**	PIR		0.96	**15**	ETS1	−0.95		**35**	STK17A		−0.94
**16**	SDC1		0.97	**36**		GSTP1	0.96	**56**	PPDPF		0.96	**16**	OSBP2	−0.95		**36**	RPS12		−0.94
**17**	FABP5		0.97	**37**		ATP6V1F	0.96	**57**	LAD1		0.96	**17**	OAF	−0.95		**37**	NPM1		−0.94
**18**	MAL		0.97	**38**		AGR2	0.96	**58**	ALDH1A1		0.96	**18**	MEIS3	−0.95		**38**	RPL18		−0.94
**19**	CAPS		0.97	**39**		CAPN1	0.96	**59**	UQCR11		0.96	**19**	COQ8B	−0.95		**39**	RPS6		−0.94
**20**	MUC15		0.97	**40**		ANXA7	0.96	**60**	CTSD		0.96	**20**	RAB31	−0.95		**40**	CFL1		−0.94
MRPL33		ALDH1A1			MGARP		ADIRF			DNAJB1	PMVK		ERICH5		ASAH1			GYG1	
ANP32A		TSTD1			TP53I3		FA2H			DAPL1	SLC20A1		FAM114A1		CAPN1			UQCR10	
UPK1B		PNKD			ATP5IF1		PSMB8			SNCG	PIR		NMRK1		TPD52			OSTF1	
HAGH		SCIN			ATP6V1F		LAMTOR2			TOB1	SYT8		WNT4		VPS28			RB1	
VAMP8		GLOD4			EML2		MAL			MRPS34	DBP		KRT3		COX5B			COA3	
PYGL		DBI			MFSD4A		SNX3			TCEA3	ARHGAP12		ATF3						

## Data Availability

Data are contained within the article and Appendix A.

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
