# Peer review of "A Keratin 12 Expression-Based Analysis of Stem-Precursor Cells and Differentiation in the Limbal–Corneal Epithelium Using Single-Cell RNA-Seq Data"

_biology, 2024, doi:10.3390/biology13030145_

Round 1
Reviewer 1 Report
Comments and Suggestions for Authors
As attached

Comments on the Quality of English LanguageQuality of English is good but could do with a careful proof reading for spelling, grammar, use of acronyms etc.
Author Response
Scientific/Technical Comments
Methodology
It is reasonably technically difficult to reproducibly dissect the same specific areas of a cornea across samples, particularly distinguishing the regions the author termed ‘LiPe’ from the ‘Pe’ which are seemingly very close together based on figure 1. The method of dissection is not well described – did the author use different diameter biopsy punches or scalpel? The former being more easily reproducible than the latter. The conformation that these regions could be reproducibly dissected (i.e. with Krt12 staining) through flow cytometry or immunohisto or cytochemical staining would have provided valuable insight into this. This is particularly pertinent in the 2 samples that were only used to isolate the ‘LiPe’ region, as they could include regions which overlap the adjacent regions as taken from single the donor where CNJ, LiPe, Pe and Co were taken. That said human tissue is often difficult to get and therefore requirements on samples can be restrictive.
Author.
Thank you for pointing to the weaknesses in the description of our protocol.
We have now included a very detailed description of how the samples were procured:
“….Cell preparation.
Corneas were radially split into 8 segments on a cutting board. Each octile was placed on the stage of a stereoscope stage fitted with a rotatable black plastic board and illuminated by a 150W Fiber Optic Dual Gooseneck Illuminator (Cole Palmer, Vernon Hills, IL). After accommodating the angle of the two illuminating beans it was possible to visualize any remaining conjunctival tissue and the limbal zone. A drop of Trypan blue was then added for 30 sec and washed with saline. The stain highlighted the freely accessible conjunctival stroma and its underlying sclera and any area of damaged or missing corneal epithelium. The conjunctiva was removed by picking up the loose tissue and cutting it out with a fine iris scissor to the edge of the scleral stain (CNJ sample, Fig.1). Next, after cutting away and discarding the blue stained sclera, a limbal-peripheral strip (LiPe sample) was collected by a cut in the peripheral zone which resulted in a width of the peripheral tissue as close as possible to the visible width of the limbus at each particular octile. Finally, cuts were made to collect a section of the adjacent periphery (Pe sample) of a width similar to the width of the periphery included in the LiPe strip, and a segment of the central cornea showing an undamaged overlying epithelium (Co sample). The surgery was performed using the tip of Extra Keen Blue single-edged blades. The octile strips from each sample type were incubated with 2 mg/ml Dispase II (Fisher, Waltham, MA) made in a bicarbonate-free 1:1 mix of Dulbecco’s modified minimum essential medium and Ham F-12 (DMEM-F12; Fisher) for 18 h, at 4° C under a 60 tilting/min motion. At the end of this treatment, sheets of epithelial cells were either free-floating in the Dispase II solution or were lightly attached to the stroma, from which they were released by gently prodding with the tip of a jeweler’s forceps. The sheets were incubated in a 0.25 % Trypsin (Fisher) solution at 37° C for 5 min (2 ml/sample), the Trypsin medium was admixed with 4 volumes of DMEM-F12 – 10 % fetal bovine serum(Atlanta Biologics , Flowery Branch, GA) and single-cell suspensions were generated by 40 passes through a 5 ml pipette.
Cells were spun down in a clinical centrifuge and resuspended, in 2 ml of DMEM/F12, triturated again, filtered through a 70 µ filter, and cultured for 3 ½ h in a 25 ml culture flask. After visually confirming the presence of single attached cells displaying side blebs indicating incipient attachment and spreading, the flask was set horizontally for 3-4 min to allow full draining of unattached cells. The accumulated medium was fully aspirated and the lightly attached cells were released by gentle streaming from a 1 ml pipette tip. The protocol recovered about ¼ of the cells in the suspension. This brief cell adhesion protocol harvested most basal epithelial cells, while excluding or drastically minimizing the recovery of supra-basal epithelial cells.
Eighty percent of the recovered cells were used for scRNA-Seq analysis. One full set of CNJ, LiPe, Pe, and Co samples (Exp. 1) were derived from a single donor and two additional LiPe samples (Exp. 2 and 3) from another two corneas. The remaining twenty percent of the samples were complemented with 1 µg/ml propidium iodide (PI) and the forward light scattering (FSC), a relative measure of overall cell size, of the PI-negative cells was used to determine the fraction of Li and Pe derived cells within the population. ….”
I hope that this description allies your concern. Also, please note that the author has been performing limbal dissections manually in this manner for over 25 years and that the percent of cells belonging to the limbal and peripheral region was accurately determined according by cell size flow cytometry, as described in the provided reference. The key to this protocol is the possibility to accurately visualize the limbus. This is possible due to several visual markers, including a) the presence of blood injected capillaries; b) a degree of turbidity of the underlying corneal-sclera zone; c) the presence of visible undulation/palisades underlying the limbus. Regarding the use of trephines, dissection will have to be done without the benefit of high magnification visualization and considering the fact that the human cornea is not perfectly circular and the width of the limbus differs from quadrant to quadrant, it may result in less accurate dissections for the purpose of this study.
Figure 1A & B
The drawing of the eye in Figure 1A does not reflect the cornea well. The arrow pointing to the central cornea actually points to what others might consider peripheral rather than the actual centre. Linked to this figure 1B I believe is perhaps trying to somehow reflect the colours used in 1A but does not really do so. More consideration should be given to this opening figure.
Author.
Thank you for pointing out to gaffes in this figure.
In response, we have made the following modifications.
- The arrow for central cornea has been extended.
- The panels B and C have now been exchanged and the legend to the figure and the narrative in the main text are properly matched. The legend for the new C now reads:
“Graphical representation of the four epithelial samples subjected to the scRNA Seq measurement. D0-D4 represent the five sub-sections within the LiPe population identified by the changes is Krt12 expression. The intensity of the red color and the size of each domains has been drawn as qualitative representation of the Krt12 level and size of each domain, respectively. The potential presence of non-epithelial cells, namely melanocytes and blood derived cells within the conjunctival and LiPe domains is indicated. …”
- Referrals to Fig. 1B or Fig 1C has been removed from Methods and are now made in the Results section only. Fig.1B is addressed in the Results first paragraph; the studies that justify the use of FSC to distinguish basal cells over the limbal surface from basal cells over the corneal surface or the supra-basal cells in the limbus are cited. Fig 1C, is addressed in the Domain Selection subsection by stating,
“, the CNJ, LiPe, and its subdomains, Pe and Co domains are graphically depicted in Fig. 1C, where the red color intensity, or lack of it, qualitatively represents the levels of Krt12 expression in the domain. “
Section 3.1 and Figure 1C
LiPe Cell analysis seems to demonstrate 2 distinct populations of cells however further characterization of these cells to confirm their type would be preferable.
Author.
I agree with this statement. However, this seems to be a task beyond the scope of this article. It will be necessary to a) identify a surface marker that undergoes a major change at the D0-D1, b) identify/generate an antibody useful for live cell flow cytometry, and c) use it to sort the two populations and perform very accurate PCR measurements, because while the statistical analysis provides a robust prove of the changes in gene expression, the increases or decreases are under two fold.
scRNA-Seq
It would be useful if the author added the read depth/sample or reads/cell they were aiming for, as well as any extra quality control measures they used beyond removing cells with low/high gene count (e.g. high numbers of UMIs/high mitochondrial read %), as well as how many reads were confidently mapped to the genome.
Author.
The parameters you mention are automatically generated by the Illumina 10x software. A cogent judging of their significance requires an in depth understanding of the 10x software process; It is not naturally evident how the specific test that may be essential for the 10x software processing will impact the processing that was performed, e.g., it may be that the trimming of outlier performed may be equivalent. Furthermore, once the data has been converted into the txt environment other readily accessible tests can be performed. Thus, in response to your query, I have introduced the following modifications. A. The genomic coverage reported by the quality report is mentioned within the Single cell RNA Seq section (“ The genomics coverage ranged between 94.2 and 96.1 % ..”). B. The subject tis re-addressed in the new Quality controls section (“ Concerning genomic coverage, the LiPe-1 sample of the scRNA Seq protocol identified 21,944 genes, or 65 percent of the 33,531 probed genes. This percentage is within the percent of probed genes displayed by a wide variety of organ and cell lines [26].”). C. The new Quality control section addresses the issue of cell health across the cell range by focusing on mitochondria and ribosomes. The mitochondrial analysis seems to be fully equivalent to the analysis that Seurat performs.
“…. however it would seem like a better strategy would be to use some kind of pseudotime analysis rather than lots of pairwise comparisons of D0/D1 etc, as this would then remove some of the issues around different size groups and “intermingling” cells, as removing these seems counterintuitive, given that their focus.
Author.
I am are not familiar with this approach. However, form what I have read it would look that it is somewhat equivalent to the correlation analysis (CA) we included in the article for the Co section. It would be great if it has been possible to use correlation for the study of the transition zone. However, the very low expression range (> 1 % of max) for Krt12 within the limbal zone coupled to the high variability from cell to cells (it is how the scRNA Seq looks, not the authors fault!) preempted the application of CA to the limbal cells. I clearly discuss the limitations of the study (e.g., Figure 5) and the need for a much larger cell base. A pseudotime approach would have to be developed and applied before it can be concluded that can overcomes the limitations of the current data.
Regarding the gaps, the text was incorrect (no need of a gap between D0 and D1, since the domain are defined by the abrupt change from Krt12 = 0 to Krt12 ≠ 0). The text says now: “ .. with the following alterations. First, in the D1-D2 and D2-D3 comparisons, the 10 last and first 10 cells of each domain were excluded to avoid any difference degrading the effect of a zone where the two domains may intermingle. A D4i.1 domain was defined as the first 1000 cells of D4i, after the exclusion of the first 100 cells post-D3 domain.”
Other modifications or additions:
- The text has been carefully reviewed for typos and poor grammar
- The text does not longer contain the words “we’, “our” or “us”
- The Tissue procurement section clearly explains why in the US the research performed does not constitute Human subject research.
- In the DISCUSSION the following paragraphs are either de novo additions or substantially modifications:
“An interesting global feature of the Krt12-linked GE changes can be gleaned from the ratio between the down to upregulated genes in the various interdomain comparisons (Table 3). Within the LiPe sample, the ratio decreases as differentiation proceeds, equaling 1.68, 1.47, and 1.11 for the D0-D123, D123-D4i.1 and D4i.1-D4i.2 comparisons. Since the number of transcripts is the same for all cells in the normalized data, the implication is that gene diversity is been progressively lost as differentiation progresses, in particular within the limbal zone itself. The pattern, though, reverts at the center of the cornea, probably because the downregulation has already been mostly completed there, as suggested by the GE pattern described in Fig. 7.
“The smaller D2 and D3 domains showed very little difference with the D1 domain, they seem to belong to cells that are mostly unchanged in gene complexation from D1. However, the subtle upsurge in these cells of the high expression genes that undergo strong increases as cell undergo a frank transition from the limbal to the periphery zones, indicates that, while still within the limbus by the Krt12 expression criterium, the cells are starting to undergo changes towards the peripheral-corneal phenotype. It is likely that had the cell number available had been larger more genes will have fallen within the BHp < 0.01 which defines significant difference in this study. To identify the genes associated with the cell transition from the vascular limbus to the avascular cornea we compared the whole D123 set against D4i.1, the first 1000 cells of D4 after exclusion of the 100 cells in the transition zone. “
“Two clustering-based studies of scRNA-Seq data found that the top downregulated gene in the D4i.1 to D4i.2 comparison, GPHA2, is acutely localized to a less differentiated subset of limbal cells [21,22]. In this study, though, GPHA2 was substantially expressed in the lower subdomain of the Pe. Furthermore, a D0-D4 plot for GPHA2 showed a distribution similar to that shown in Fig. 4 for Krt75. Since Krt75 belongs in the same Table 8 list as GPHA2, it was intriguing to examine the distribution of these two genes in the 20 quantiles of D4 used in Fig. 6 and the relationship to other downregulating genes of the D4i.1-D4i.2 comparison. Fig. 7 describes the changes in GPHA2 and for the 4 genes with the highest correlation coefficient (> 0.95) to its distribution. The inset shows the combination of the previously defined D0-D3 domains with the first 4 quantiles of D4 for GPHA2 Krt12 and Krt3. It is clear that in our analysis GPHA2 achieves its maximum expression at the very start of the cell transition to the Pe domain (D4Q1). The other genes in Fig. 7 display a very similar domain distribution pattern (not shown). The apparent discrepancy between both approaches remains to be resolved.”
Reviewer 2 Report
Comments and Suggestions for Authors
The manuscript by J. Mario Wolosin describes a single cell RNA seq approach to K12 expression-based analysis of various limbal stem cell precursors and their role in corneal epithelial differentiation. The study is well designed with interesting data. However, some concerns remain:
1. The authors have not performed any validation studies to the single cell data at gene (qPCR) or protein (immunostaining) level to support the bioinformatics analysis obtained from the single cell experiment.
2. Minor concern: copy editing for English language grammar and typos should be performed.
Comments on the Quality of English LanguageMinor editing for English language grammar and typos is required.
Author Response
The manuscript by J. Mario Wolosin describes a single cell RNA seq approach to K12 expression-based analysis of various limbal stem cell precursors and their role in corneal epithelial differentiation. The study is well designed with interesting data. However, some concerns remain:
- The authors have not performed any validation studies to the single cell data at gene (qPCR) or protein (immunostaining) level to support the bioinformatics analysis obtained from the single cell experiment.
Author
Unfortunately, I am not presently able to perform PCR confirmation. Instead we have added a Quality controls and validations section showing that the data conforms to previously established relationship between differentiation and gene expression. It seems logical that if the data confirms previous known facts for marker genes is highly likely to be true for any previously unknown trend. A full quality control section has been added.
Other modifications or additions:
- The text has been carefully reviewed for typos and poor grammar
- The text does not longer contain the words “we’, “our” or “us”
- The Tissue procurement section clearly explains why in the US the research performed it does not constitute Human subject research.
- In the DISCUSSION the following paragraphs are either de novo additions or substantially modifications:
“An interesting global feature of the Krt12-linked GE changes can be gleaned from the ratio between the down to upregulated genes in the various interdomain comparisons (Table 3). Within the LiPe sample, the ratio decreases as differentiation proceeds, equaling 1.68, 1.47, and 1.11 for the D0-D123, D123-D4i.1 and D4i.1-D4i.2 comparisons. Since the number of transcripts is the same for all cells in the normalized data, the implication is that gene diversity is been progressively lost as differentiation progresses, in particular within the limbal zone itself. The pattern, though, reverts at the center of the cornea, probably because the downregulation has already been mostly completed there, as suggested by the GE pattern described in Fig. 7.
“The smaller D2 and D3 domains showed very little difference with the D1 domain, they seem to belong to cells that are mostly unchanged in gene complexation from D1. However, the subtle upsurge in these cells of the high expression genes that undergo strong increases as cell undergo a frank transition from the limbal to the periphery zones, indicates that, while still within the limbus by the Krt12 expression criterium, the cells are starting to undergo changes towards the peripheral-corneal phenotype. It is likely that had the cell number available had been larger more genes will have fallen within the BHp < 0.01 that defines significant difference in this study. To identify the genes associated with the cell transition from the vascular limbus to the avascular cornea we compared the whole D123 set against D4i.1, the first 1000 cells of D4 after exclusion of the 100 cells in the transition zone. “
“Two clustering-based studies of scRNA-Seq data found that the top downregulated gene in the D4i.1 to D4i.2 comparison, GPHA2, is acutely localized to a less differentiated subset of limbal cells [21,22]. In this study, though, GPHA2 was substantially expressed in the lower subdomain of the Pe. Furthermore, a D0-D4 plot for GPHA2 showed a distribution similar to that shown in Fig. 4 for Krt75. Since Krt75 belongs in the same Table 8 list as GPHA2, it was intriguing to examine the distribution of these two genes in the 20 quantiles of D4 used in Fig. 6 and the relationship to other downregulating genes of the D4i.1-D4i.2 comparison. Fig. 7 describes the changes in GPHA2 and for the 4 genes with the highest correlation coefficient (> 0.95) to its distribution. The inset shows the combination of the previously defined D0-D3 domains with the first 4 quantiles of D4 for GPHA2 Krt12 and Krt3. It is clear that in our analysis GPHA2 achieves its maximum expression at the very start of the cell transition to the Pe domain (D4Q1). The other genes in Fig. 7 display a very similar domain distribution pattern (not shown). The apparent discrepancy between both approaches remains to be resolved.”
Reviewer 3 Report
Comments and Suggestions for Authors
This is a novel study aimed at identifying gene expression changes that coincide with k12 expression changes in the cornea epithelium. The group is highly experienced in this area. I have only minor comments.
It would be helpful to know if the stem cell cluster gene expression was consistent with previously published negative (e.g. Cx43) and positive (e.g. N cadherin)
Did they notice any changes to the expression of pax6 which is known to regulate k12?
Author Response
This is a novel study aimed at identifying gene expression changes that coincide with k12 expression changes in the cornea epithelium. The group is highly experienced in this area. I have only minor comments.
It would be helpful to know if the stem cell cluster gene expression was consistent with previously published negative (e.g. Cx43) and positive (e.g. N cadherin)
Did they notice any changes to the expression of pax6 which is known to regulate k12?
Author
We have added an extensive Quality control and validation section. It includes both GJA1/Cx43 (please note that the claim as to Cx43 has been that it is express at substantial lower level protein in the limbus with a small fraction of the cells are fully negative for protein) and PAX6. CDH2 expression is below our threshold expression for reliable data.
Other modifications or additions:
- The text has been carefully reviewed for typos and poor grammar
- The text does not longer contain the words “we’, “our” or “us”
- The tissue procurement section clearly explains why in the US the research performed does not constitute Human subject research.
- In the DISCUSSION the following paragraphs are either de novo additions or substantially modifications:
“An interesting global feature of the Krt12-linked GE changes can be gleaned from the ratio between the down to upregulated genes in the various interdomain comparisons (Table 3). Within the LiPe sample, the ratio decreases as differentiation proceeds, equaling 1.68, 1.47, and 1.11 for the D0-D123, D123-D4i.1 and D4i.1-D4i.2 comparisons. Since the number of transcripts is the same for all cells in the normalized data, the implication is that gene diversity is been progressively lost as differentiation progresses, in particular within the limbal zone itself. The pattern, though, reverts at the center of the cornea, probably because the downregulation has already been mostly completed there, as suggested by the GE pattern described in Fig. 7.
“The smaller D2 and D3 domains showed very little difference with the D1 domain, they seem to belong to cells that are mostly unchanged in gene complexation from D1. However, the subtle upsurge in these cells of the high expression genes that undergo strong increases as cell undergo a frank transition from the limbal to the periphery zones, indicates that, while still within the limbus by the Krt12 expression criterium, the cells are starting to undergo changes towards the peripheral-corneal phenotype. It is likely that had the cell number available had been larger more genes will have fallen within the BHp < 0.01 that defines significant difference in this study. To identify the genes associated with the cell transition from the vascular limbus to the avascular cornea we compared the whole D123 set against D4i.1, the first 1000 cells of D4 after exclusion of the 100 cells in the transition zone. “
“Two clustering-based studies of scRNA-Seq data found that the top downregulated gene in the D4i.1 to D4i.2 comparison, GPHA2, is acutely localized to a less differentiated subset of limbal cells [21,22]. In this study, though, GPHA2 was substantially expressed in the lower subdomain of the Pe. Furthermore, a D0-D4 plot for GPHA2 showed a distribution similar to that shown in Fig. 4 for Krt75. Since Krt75 belongs in the same Table 8 list as GPHA2, it was intriguing to examine the distribution of these two genes in the 20 quantiles of D4 used in Fig. 6 and the relationship to other downregulating genes of the D4i.1-D4i.2 comparison. Fig. 7 describes the changes in GPHA2 and for the 4 genes with the highest correlation coefficient (> 0.95) to its distribution. The inset shows the combination of the previously defined D0-D3 domains with the first 4 quantiles of D4 for GPHA2 Krt12 and Krt3. It is clear that in our analysis GPHA2 achieves its maximum expression at the very start of the cell transition to the Pe domain (D4Q1). The other genes in Fig. 7 display a very similar domain distribution pattern (not shown). The apparent discrepancy between both approaches remains to be resolved.”
Round 2
Reviewer 2 Report
Comments and Suggestions for Authors
The reviewer has no further comments.